

# Exact Mazur bounds in the pair-flip model and beyond

Oliver Hart⋆

Department of Physics and Center for Theory of Quantum Matter,
University of Colorado Boulder, Boulder, Colorado 80309 USA

⋆ oliver.hart-1@colorado.edu

## Abstract

By mapping the calculation of Mazur bounds to the enumeration of walks on fractal structures, we present exact bounds on the late-time behavior of spin autocorrelation functions in models exhibiting pair-flip dynamics and more general $p$-flip dynamics. While the pair-flip model is known to exhibit strong Hilbert space fragmentation, the effect of its nontrivial conservation laws on autocorrelation functions has, thus far, only been calculated numerically, which has led to incorrect conclusions about their thermodynamic behavior. Here, using exact results, we prove that infinite-temperature autocorrelation functions exhibit infinite coherence times at the boundary, and that bulk Mazur bounds decay asymptotically as $1/\sqrt{L}$, rather than $1/L$, as had previously been thought. This result implies that the nontrivial conserved operators implied by $p$-flip dynamics have an important *qualitative* impact on bulk thermalization properties beyond the constraints imposed by the simple global symmetries of the models.

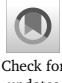

# 1 Introduction

Conserved quantities play a crucial role in our understanding of many-body systems and place strong constraints on their dynamics. It is widely established that all isolated many-body quantum systems possess a number of conserved quantities that is exponential in system volume: projectors onto their eigenstates. However, the generic expectation is that these projectors are highly nonlocal. Hence, they have little influence on the thermalization of local observables in which we are typically interested, a statement that is formalized by the eigenstate thermalization hypothesis (ETH) [1,2]. Recently, there has been a renewed interest in understanding the situations wherein this expectation breaks down. The extensive number of *local* conserved quantities possessed by (Yang-Baxter) integrable systems [3,4] endow them with local memory of their initial state and hence prevent such systems from thermalizing to any conventional ensemble from statistical mechanics. Similarly, it was realized that the mechanism of many-body localization (MBL) [5,6] can also give rise to emergent (quasi-) local conserved quantities – without fine tuning – in the presence of sufficiently strong disorder. More recently, a number of novel mechanisms that preclude thermalization in translation-invariant systems have been proposed. These include systems with quantum many-body scars (QMBS) [7–11], whose spectra contain a small number of ETH-violating states, and *Hilbert space fragmentation* (aka shattering) [12,13], where the Hamiltonian factorizes into a direct sum of exponentially many dynamically disconnected (i.e., Krylov) sectors when expressed in a particular local basis, among others [14–18]. Correspondingly, such fragmented systems posses exponentially many conserved quantities that are intermediate between eigenstate projectors and global symmetries; broadly, this manuscript deals with quantifying the constraints that these conserved quantities place on thermalization.

Following the discovery of ergodicity breaking in random unitary circuits that preserve charge and its first (i.e., dipole) moment [19], later explained in terms of Hilbert space fragmentation [12,13], there has been a great deal of interest in the interplay between quantum dynamics and multipolar symmetries [20–45]. It is now understood that Hilbert space fragmentation can arise in more generic settings from requiring that the system's Hamiltonian both respects certain symmetries and obeys strict spatial locality (although it has recently been shown that the latter is not necessary when fragmentation is protected by higher-form symmetries [46,47]), and comes in two flavors, weak and strong, depending on the distri-

bution of Krylov sector dimensions. Weakly fragmented systems share many features with QMBS [48]; there is one dominant Krylov sector that plays the role of the thermalizing bulk, and a measure-zero set of ETH-violating states. Strongly fragmented systems, on the other hand, do not possess a dominant Krylov sector. Therefore, even dynamics from typical infinite-temperature states can exhibit fingerprints of the system's initial conditions in the form of infinite autocorrelation times, as diagnosed by infinite-temperature autocorrelation functions for local observables.

More generally, infinite-temperature autocorrelation functions $A_O(t) = \langle \hat{O}(t)\hat{O}(0)\rangle_{\beta=0}$, for some local operator $\hat{O}$, are an invaluable tool for diagnosing thermalization properties of many-body quantum systems. In weakly fragmented systems, the late-time behavior is indicative of the nature of transport in the system. For instance, in models with multipole moment conservation, the late-time behavior of correlation functions is described by "fracton hydrodynamics" [27], and the subdiffusive transport therein can be inferred from the power-law decay of $A_O(t)$ at late times (see, e.g., Refs. [36–45]). Since typical states have vanishing overlap with any ETH-violating states, infinite-temperature correlation functions are not sensitive to weak fragmentation. Conversely, in strongly fragmented systems, it is possible to obtain *operator localization* [12, 13, 19] for strictly local operators, even at infinite temperature. If the late-time average of $A_O(t)$ remains nonzero in the thermodynamic limit, this implies that the Heisenberg-evolved operator $\hat{O}(t)$ exhibits nonzero overlap with its initial condition at all times. This occurs in, e.g., spin-1 dipole-conserving models with sufficiently short-ranged Hamiltonians [12, 13]. The late-time behavior of $A_O(t)$ can also capture the more subtle effects of fragmentation in, e.g., the $t$-$J_z$ model [20]. There, operators at the boundary remain localized, reminiscent of strong zero modes [49–54], while (the late-time averages of) bulk autocorrelation functions decay with system size as $1/\sqrt{L}$, in a one-dimensional (1D) system of size $L$. In all cases, the late-time behavior of such autocorrelation functions can be quantified analytically using Mazur bounds [55–57], which project $\hat{O}$ onto a set of conserved operators. This approach has been applied successfully to bound tightly the late-time behavior of infinite-temperature autocorrelation functions in a variety of models exhibiting fragmentation [12, 20, 24, 58].

This manuscript is concerned with the behavior of autocorrelation functions in the pair-flip model [24, 47, 59, 60], and its generalization to $p$-flip dynamics, as diagnosed by Mazur bounds. These models describe dynamics in which spins are only dynamical if they belong to a group of $p$ contiguous spins, each of which is in the same state, in which case the entire group of $p$ spins must flip together. This dynamics gives rise to a large number of conserved quantities that derive from conserved "patterns" of spins, encoded throughout the system nonlocally. These conserved quantities lead to strong fragmentation. The conservation of a nonlocally encoded pattern, obtained via some local real-space decimation procedure, defines a family of fragmented models, including the $t$-$J_z$ model [20] and the model of Ref. [22]. One might therefore expect such models to exhibit similar thermalization properties. However, it has been conjectured [24] (and substantiated numerically [60]) that, at late times, autocorrelation functions in the pair-flip model exhibit operator localization at the boundary but decay as $1/L$ in the bulk. This bulk behavior is typical of generic, nonfragmented systems that preserve only a U(1) symmetry, and would suggest that the spin patterns conserved by the dynamics do not change the scaling that one obtains from the model's continuous symmetries. Consequently, there would be no notion of infinite-temperature operator localization in the bulk implied by the model's strong fragmentation. This behavior would be surprising, and suggests that the dynamics of the pair-flip model is qualitatively different from other canonical models of fragmentation. To date, an understanding of this alleged difference in behavior has remained an open question.

We present an exact mapping between Mazur bounds in $p$-flip models and the enumeration of constrained walks on specific fractal structures. Using this mapping, we show that the late-time averages of infinite-temperature autocorrelation functions saturate to a nonzero value at the boundary, but decay asymptotically as $1/\sqrt{L}$ in the bulk, contrary to the conjecture of Ref. [24]. This result implies that the strong fragmentation of the model *does* have a qualitative impact on operator localization, *even in the bulk*, where delocalization occurs only over a distance $\sqrt{L} \ll L$ by virtue of the model's conserved patterns. Hence, this manuscript relieves the conceptual tension that previously existed between the pair-flip model and other models exhibiting strong fragmentation. Furthermore, we provide an intuitive explanation of the model's boundary localization: using the formalism of Ref. [20], we write down a conserved operator that is "localized" (see Refs. [20, 61]) at the edge of the system.

The manuscript is structured as follows. Models exhibiting $p$-flip dynamics are introduced in Sec. 2, including a discussion of their protecting symmetries and their numerous conserved quantities, which lead to Hilbert space fragmentation. Next, in Sec. 3, we discuss the impact of various conserved quantities on infinite-temperature autocorrelation functions in the context of Mazur bounds. In particular, the contribution from the models' continuous symmetries is discussed, and the contribution from *all* conserved operators implied by the conserved pattern is formally written down. The section closes with a discussion of why an exact solution is necessary. A nontechnical summary of the main accomplishments of the manuscript is then presented in Sec. 4.1. In particular, after presenting the mapping to enumeration of walks on fractal structures, we present the Mazur bounds at the boundary and in the bulk of the system, and provide an intuitive physical interpretation with supporting numerical results. For those readers interested in the technical details pertaining to the exact calculations, they are presented in Secs. 4.2–4.4. Finally, a discussion of the results is presented in Sec. 5.

# 2 Models with $p$-flip dynamics

## 2.1 Symmetries and dynamics

Consider a one-dimensional lattice composed of $m$-level systems (spins) on vertices, spanned by states $|\alpha\rangle \in \{|0\rangle, \ldots, |m-1\rangle\}$, which will represent different colors. Motivated by recent work on the pair-flip (PF) and related models [24, 47, 59, 60, 62], we consider a generalized class of dynamics that conserves the following U(1) symmetry charges:

$$\hat{N}^\alpha = \sum_{j=0}^{L-1} \exp\left(\frac{2\pi \mathrm{i} j}{p}\right) \hat{P}_j^\alpha, \tag{1}$$

where $\hat{P}_j^\alpha = |\alpha\rangle\langle\alpha|_j$ is the projector onto state (color) $\alpha$ on site $j$, and $p$ is a prime number.[1] Note that only $m-1$ of the operators $\hat{N}^\alpha$ are independent since the projectors satisfy $\sum_{\alpha=0}^{m-1} \hat{P}_j^\alpha = \mathbb{1}$ on each site. The most local dynamics compatible with the symmetries (1) is as follows: If there exists a run of $p$ contiguous, identically colored spins, say color $\alpha$, these $p$ spins can simultaneously be cycled to another state, say $\beta$, with $0 \le \alpha, \beta \le m-1$. We refer to

---

[1] If $p$ is not prime, then we can instead enforce $\hat{N}^\alpha = \sum_{j=0}^{L-1} \exp\left(\frac{2\pi \mathrm{i} j q}{p}\right) \hat{P}_j^\alpha$ for all integers $q$ that divide $p$ satisfying $1 \le q < p$. This ensures that no subset of spins belonging to a $p$-site unit cell can be flipped while preserving the symmetries. However, this set of symmetries is, in general, sufficient but not necessary to enforce $p$-flip dynamics.

these local updates as "$p$-flip" dynamics, generated by the Hamiltonian[2]

$$\hat{H}_{p,m} = -\sum_i \sum_{\alpha,\beta} g_i^{\alpha\beta} |\alpha\alpha\cdots\alpha\rangle\langle\beta\beta\cdots\beta|_{[i,i+p)} - \sum_i \sum_\alpha \lambda_i^\alpha |\alpha\rangle\langle\alpha|_i . \tag{2}$$

The coefficients $g_i^{\alpha\beta}$ and $\lambda_i^\alpha$ are arbitrary (subject to Hermiticity requirements), and serve to break all discrete or continuous symmetries that the model may otherwise possess. As a result, the model might be expected to exhibit polynomially many[3] distinct symmetry sectors based on the U(1) symmetries enforced by (1). However, as we will now show, the Hamiltonian (2) instead "shatters" [12,13] into *exponentially* many Krylov sectors, which remain disconnected under the dynamics generated by (2). Note that the Hamiltonian (2) may be transformed [47] via a Kramers-Wannier duality to a PXP-like [9] model, connecting it to models with facilitated spin-flip dynamics [66,67].

## 2.2 Conserved patterns and Krylov sectors

In the following discussion, unless otherwise stated, we consider $m = 3$ colors and triple-flip dynamics ($p = 3$) for convenience of illustration, but the concepts are readily generalized to other numbers of colors and to general $p$-flip dynamics (see Refs. [24,47] for a detailed discussion of pair-flip dynamics). Further, we will exclusively consider systems with open boundaries, since we will be interested in quantifying boundary effects in later sections. Introducing the graphical representation of states

$$|0\rangle \equiv |\textcolor{red}{\bullet}\rangle , \qquad |1\rangle \equiv |\textcolor{green}{\bullet}\rangle , \qquad |2\rangle \equiv |\textcolor{blue}{\bullet}\rangle , \tag{3}$$

we can denote "active" triplets of spins, which can directly be permuted to another color by (2), by joining up their legs:

$$|\textcolor{red}{\bullet}\ \textcolor{red}{\bullet}\ \textcolor{red}{\bullet}\rangle \rightarrow |\overset{\frown}{\textcolor{red}{\bullet\!-\!\bullet\!-\!\bullet}}\rangle . \tag{4}$$

We refer to the combination of grouped spins on the right-hand side as a "trimer." Suppose that we perform this grouping procedure (from left to right, say) on a particular spin configuration, and subsequently remove the grouped spins from the state, which may produce new neighbors with the same color. This grouping and removal ("decimation") procedure can then be repeated until we are left with a spin configuration that contains no runs of any color of length three (or greater), i.e., the spins that remain at the end of the procedure cannot be grouped without introducing crossings between the shaded regions. Hence, the decimation procedure produces a configuration of ungrouped dots separated by noncrossing, grouped trimers. As an explicit example of this decimation procedure:

$$\left|\textcolor{blue}{\bullet}\ \overset{\frown}{\textcolor{red}{\bullet}\,\textcolor{green}{\bullet\!-\!\bullet}\,\textcolor{red}{\bullet}}\ \textcolor{green}{\bullet}\ \textcolor{green}{\bullet}\ \overset{\frown}{\textcolor{blue}{\bullet\!-\!\bullet\!-\!\bullet}}\right\rangle \mapsto |\textcolor{blue}{\bullet}\ \textcolor{green}{\bullet}\ \textcolor{green}{\bullet}\rangle . \tag{5}$$

The pattern of dots on the right-hand side corresponds to a conserved quantity for dynamics generated by (2). To see this, first observe that any contiguous region of $3n$ spins grouped into noncrossing shaded trimers (which will be removed by the decimation procedure) can be connected by triple-flip dynamics. For example, under (2),

$$\left|\overset{\frown}{\textcolor{red}{\bullet}\,\textcolor{green}{\bullet\!-\!\bullet}\,\textcolor{red}{\bullet}}\right\rangle \leftrightarrow \left|\overset{\frown}{\textcolor{red}{\bullet\!-\!\bullet}}\ \overset{\frown}{\textcolor{green}{\bullet\!-\!\bullet}}\right\rangle . \tag{6}$$

---

[2]While we choose to present our results in terms of a time-independent Hamiltonian, the conserved quantities that we introduce also apply to random unitary circuits that implement discrete "$p$-flip" dynamics. Analogous classical models have also been considered in the context of deposition-evaporation phenomena [63–65].

[3]For the pair-flip model, there are $m-1$ independent U(1) symmetries leading to $O(L^{m-1})$ symmetry sectors. For the triple-flip model and prime $p > 3$, we instead obtain $O(L^{2(m-1)})$ symmetry sectors, since (1) contains two linearly independent charges for each color (e.g., the real and imaginary parts).

To connect the configuration on the left-hand side to the configuration on the right-hand side, the central green trimer can be flipped to red. From the resulting all-red configuration, we can then flip the rightmost three spins to green to obtain the desired spin configuration. For a generic region of noncrossing trimers, we can work outwards, starting with the most nested trimer, flipping it to be the same color as its neighboring spins. This can be repeated until the region is colored uniformly.

Second, observe that any ungrouped dot can move past a region of noncrossing trimers. The region can first be "flattened" according to the procedure in (6), then, for each disconnected trimer belonging to the region,

$$|\text{\color{red}\bullet}\ \text{\color{green}\bullet\!-\!\bullet}\rangle \leftrightarrow |\text{\color{red}\bullet}\ \text{\color{red}\bullet\!-\!\bullet}\rangle \leftrightarrow |\text{\color{red}\bullet\!-\!\bullet}\ \text{\color{red}\bullet}\rangle \leftrightarrow |\text{\color{green}\bullet\!-\!\bullet}\ \text{\color{red}\bullet}\rangle. \tag{7}$$

Two dots cannot move past one another under $p$-flip dynamics, however. Therefore, the dynamics (2) conserves the pattern obtained via the decimation procedure [as in, e.g., (5)]. The resulting patterns, which correspond to a sequence of colors with a maximum run length of $p-1$, therefore label different dynamically disconnected sectors. As a result, we will refer to the pattern produced via decimation as a *label* for the corresponding Krylov sector. If the system has length $L$, then the label must have length $L-pn$ for integer $n$, since it is obtained via the removal of $p$-spin clusters. The number of labels for generic $p$ and $m$ is enumerated by the generating function (see, e.g., Sloane's A121907 and A181137)

$$G_{p,m}(x) = \frac{1 + x + x^2 + \cdots + x^{p-1}}{1 - (m-1)(x + x^2 + \cdots + x^{p-1})}. \tag{8}$$

Specifically, the number of labels of length $\ell$ is obtained by extracting the coefficient $[x^\ell]G_{p,m}(x)$, i.e., the coefficient of $x^\ell$ in the formal power series expansion of (8) around $x = 0$. For triple-flip dynamics, the number of labels scales *exponentially* with the length of the label $\ell$ for all $m \geq 2$ (and, hence, the total number of labels, found by summing over all $\ell = L - pn$, scales exponentially with system size $L$). For example, the number of labels of length $\ell \geq 1$ for $m = 2$ colors has the closed-form expression

$$[x^\ell]G_{3,2}(x) = \frac{2}{\sqrt{5}}(\varphi^{\ell+1} - (-\varphi)^{-\ell-1}) = 2F_{\ell+1}, \tag{9}$$

with $\varphi$ the golden ratio, and $F_n$ the $n$th Fibonacci number. Note that this behavior is in stark contrast to the *pair*-flip model ($p = 2$) [24, 47], in which the number of labels grows exponentially with $\ell$ only for $m \geq 3$; for $m = 2$, the pair-flip model has just two labels (obtained via an analogous decimation procedure described in Refs. [24, 47, 59]) of any given length corresponding to an alternating sequence of the two colors.

The conserved operators implied by the above discussion correspond to a generalization of the U(1) symmetries in (1). In particular, $p$-flip dynamics conserves the operators

$$\hat{N}^{\alpha_1\alpha_2\cdots\alpha_k} = \sum_{j_1 < j_2 < \cdots < j_k} \exp\left(\frac{2\pi i}{p} \sum_{n=1}^{k} j_n\right) \prod_{n=1}^{k} \hat{P}_{j_n}^{\alpha_n}, \tag{10}$$

where $\alpha_1, \alpha_2, \ldots, \alpha_k$, for $k \leq L$, corresponds to a sequence of colors with maximum run length $p-1$ (i.e., the sequence of colors forms a valid label for a system of size $k$ with open boundaries). Equation (10) generalizes the operators from Ref. [24] to $p$-flip dynamics. Depending on the length of the sequence of colors $\alpha_1, \alpha_2, \ldots, \alpha_k$, the operators (10) interpolate between the U(1) symmetries (1) (for $k = 1$) and projectors onto frozen states ($k = L$), and are generically spatially nonlocal for $k \geq 2$. Since the number of linearly independent conserved operators must equal the number of Krylov sectors [24], it is clear that not all of the operators in (10) are linearly independent; some of their interdependencies are discussed in Appendix A. Note that there can exist a more local set of conserved quantities that generate the operators in (10); this is a point to which we return in Sec. 4.1.4.

# 3 Mazur bounds

Section 2.2 illustrates that systems exhibiting $p$-flip dynamics preserve an irreducible label obtained via a real-space decimation procedure (5), and that each such label corresponds to a distinct Krylov sector. Dynamics that conserve the operators in (10) ostensibly cannot fully thermalize with respect to the U(1) symmetry sectors implied by (1). Here, we discuss how Mazur bounds [55–57] can be used to quantify the obstructions to thermalization that these conserved labels place on the system's dynamics by bounding infinite-temperature autocorrelation functions.

## 3.1 General bound

Mazur bounds have a long history and have been used extensively in the context of integrable systems, where the Mazur inequality places bounds on transport coefficients such as the Drude weight [68–74] (see also Refs. [75–77]). Typically, only a few local or quasilocal conserved quantities are considered. Here, since we will be interested the precise decay of bulk Mazur bounds with system size, it is convenient to work with the *entire* family (10). Indeed, as shown in Ref. [24], this is is required to give rise to tight bounds throughout the bulk.

   Given a set of quantities $\{\hat{I}_\mu\}$ that are conserved by a system's dynamics, $[\hat{H}, \hat{I}_\mu] = 0$ for all $\mu$, we can bound the late-time average of autocorrelation functions of some Heisenberg-evolved observable $\hat{O}(t)$ by projecting onto the set of conserved quantities:

$$C_O \equiv \lim_{\tau \to \infty} \frac{1}{\tau} \int_0^\tau dt \, \langle \hat{O}(t)\hat{O}(0) \rangle_{\beta=0} \geq \sum_{\mu\nu} (O|I_\mu)(K^{-1})_{\mu\nu}(I_\nu|O). \tag{11}$$

In this expression $\langle \cdots \rangle_{\beta=0} \equiv D^{-1} \text{Tr}(\cdots)$ is the infinite-temperature expectation value with $D$ the total dimension of the Hilbert space, and, on the right-hand side, we have utilized the operator inner product $(A|B) \equiv D^{-1} \text{Tr}(\hat{A}^\dagger \hat{B})$. The matrix $K_{\mu\nu}$ is defined by the overlap between the various conserved quantities $K_{\mu\nu} \equiv (I_\mu|I_\nu)$. If we make use of an orthonormal set of conserved quantities satisfying $(Q_\mu|Q_\nu) = \delta_{\mu\nu}$, the general bound (11) is simplified to

$$C_O \geq \sum_\mu (O|Q_\mu)(Q_\mu|O). \tag{12}$$

In what follows, we are interested in quantifying the late-time average of the spin autocorrelation functions $\langle \hat{S}_i^z(t)\hat{S}_i^z(0) \rangle_{\beta=0}$ evaluated at site $i$ of a 1D chain with open boundaries. Note that $\hat{S}_i^z$ is defined in terms of the color degrees of freedom as $\hat{S}_i^z = \sum_{\alpha=0}^{m-1} \frac{1}{2}(m-1-2\alpha)|\alpha\rangle\langle\alpha|$. Such correlation functions give insight into whether the system thermalizes, since infinite autocorrelation times imply that the system retains "memory" of its initial conditions, as is the case in many-body-localized systems [5,6], spin glasses and kinetically constrained models [66,67,78], and systems with strong zero modes [49–54], for example.

## 3.2 Bound from U(1) symmetries

First, we evaluate the bound (11) using only the U(1) symmetry charges $\hat{N}^\alpha$ (1), i.e., neglecting all of the additional conserved quantities in (10). For simplicity, we assume that the length of the chain is an integer multiple of $p$, i.e., it is $p$-partite. The conserved quantities $\hat{N}^\alpha$ are not automatically orthonormal with respect to the inner product $(N^\alpha|N^\beta)$; indeed, only $m-1$ of them are independent, since $\sum_{\alpha=0}^{m-1} \hat{N}^\alpha = 0$, and the linearly independent operators are not automatically orthogonal. Consequently, we must first construct the overlap matrix

$$K_{ab}^{\alpha\beta} \equiv (N_a^\alpha|N_b^\beta) = \frac{L}{2m}\left(\delta^{\alpha\beta} - \frac{1}{m}\right)\delta_{ab}, \tag{13}$$

where the indices $a$, $b$ run over the real and imaginary parts of (1), and $\alpha, \beta$ run over the restricted set $\alpha, \beta \in \{1, \ldots, m-1\}$, say. This ensures that the symmetry charges used for the bound are linearly independent from one another.[4] For the pair-flip model, only the real part of $\hat{N}^\alpha$ is nontrivial and there is no factor of $1/2$ in (13). Using a restricted set of colors, the matrix inverse of (13) exists and is given by

$$(K^{-1})^{\alpha\beta}_{ab} = \frac{2m}{L}\left(\delta^{\alpha\beta} + 1\right)\delta_{ab}. \tag{14}$$

Plugging this expression for the inverted overlap matrix into the generic bound (11), and utilizing the result $(S^z_j | N^\alpha) = S^z_j(\alpha)e^{2\pi ij/p}/m$ for the overlap between the spin on site $j$ and the symmetry charges, with $S^z_j(\alpha) = (m-1-2\alpha)/2$, we arrive at

$$C^z_i \geq \langle \hat{S}^z_i(0)\hat{S}^z_i(0)\rangle \times \begin{cases} \frac{1}{L}, & \text{for } p = 2, \\ \frac{2}{L}, & \text{for } p > 2. \end{cases} \tag{15}$$

The bound is therefore independent of the location of the spin being bounded, and decays with system size as $L^{-1}$. The interpretation of (15) is simple: If the system thermalizes [while conserving the U(1) charges in (1)], charge that is initially localized on site $i$ eventually spreads homogeneously throughout the system of size $L$, implying that the overlap with the initial state should scale as $L^{-1}$. Hence, if the autocorrelation function plateaus at a value parametrically larger than (15), it implies the presence of restricted thermalization (equivalently, charge is prevented from spreading homogeneously throughout the system).

## 3.3 Bound from all conserved quantities

The conserved labels identified in Sec. 2.2 imply that there are many conserved quantities beyond the U(1) symmetries protecting the fragmentation (1). Since it is not *a priori* clear which of the conserved quantities (10) will contribute most significantly to the bound, we can indiscriminately account for *all* such conserved quantities by using the projectors onto Krylov sectors, defined by the conserved labels, $\hat{P}_\mu$, as a set of mutually orthogonal (and linearly independent) conserved quantities. That is, $\hat{P}_\mu$ projects onto all spin configurations that produce the label $\mu$ upon performing the decimation procedure (5). Note that the vectorized projectors $\hat{P}_\mu \to |P_\mu)$ satisfy $(P_\mu|P_\nu) = D^{-1}\delta_{\mu\nu}\text{Tr}(P_\mu) = \delta_{\mu\nu}d_\mu/D$ (no summation), with $d_\mu$ the dimension of the Krylov sector labeled by $\mu$. Hence, we can define the normalized conserved quantities

$$|\pi_\mu) = \sqrt{\frac{D}{d_\mu}}|P_\mu), \quad \text{satisfying} \quad (\pi_\mu|\pi_\nu) = \delta_{\mu\nu}. \tag{16}$$

The simplified expression (12) can therefore be used for the normalized set of projectors $|\pi_\mu)$. Meanwhile, the overlap between $\hat{S}^z_i$ on site $i$ and the operator $\hat{\pi}_\mu$ becomes

$$(\pi_\mu|S^z_i) = \frac{1}{\sqrt{d_\mu D}}\text{Tr}(\hat{P}_\mu \hat{S}^z_i) = \frac{1}{\sqrt{d_\mu D}}\sum_{\mathbf{s}}\langle \mathbf{s}|\hat{P}_\mu \hat{S}^z_i|\mathbf{s}\rangle = \frac{1}{\sqrt{d_\mu D}}\sum_{\mathbf{s}\in\mu}\langle \mathbf{s}|\hat{S}^z_i|\mathbf{s}\rangle. \tag{17}$$

In the second equality, we expressed the trace in terms of computational basis states $|\mathbf{s}\rangle$ (upon which the action of $\hat{S}^z_i$ and $\hat{P}_\mu$ is diagonal), allowing us to restrict to states belonging to the Krylov sector labeled by $\mu$. The bound therefore simplifies to

$$C^z_i \geq \frac{1}{D}\sum_\mu \frac{1}{d_\mu}\left[\sum_{\mathbf{s}\in\mu}\langle \mathbf{s}|\hat{S}^z_i|\mathbf{s}\rangle\right]^2. \tag{18}$$

---

[4]The final result is independent of which index is excluded.

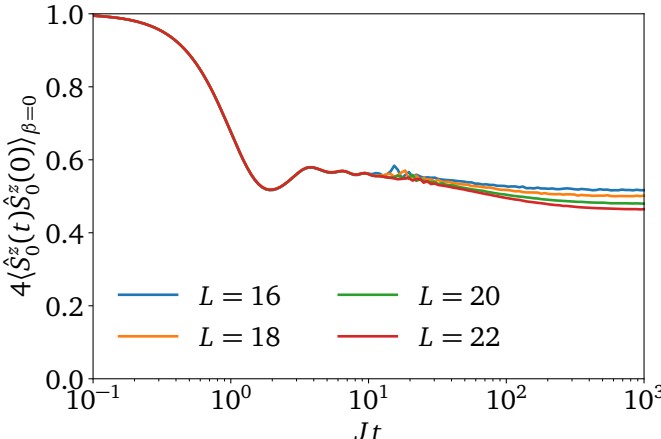

Figure 1: Numerical time evolution of the translation-invariant, spin-1/2 triple-flip model for increasing system sizes. Whether the late-time plateaux converge to a nonzero value as $L \to \infty$ cannot be reliably inferred from the data, motivating the need for an exact solution. The infinite-temperature average is performed using a stochastic approximation to the trace; the corresponding error bars are smaller than the line width in all cases.

This expression accounts for *all* conserved quantities associated to the conserved pattern, and therefore subsumes the bounds based on symmetry alone (15). To evaluate the bound (18) we therefore need to know the Krylov sector dimensions $d_\mu$ and how to evaluate expectation values of $\hat{S}_i^z$ averaged over states belonging to a particular Krylov sector. Remarkably, both of these quantities can be evaluated *exactly* by enumerating walks on specific fractal graphs, as will be explained in Sec. 4. First, we motivate why it is desirable to have such an exact solution.

## 3.4 Need for an exact solution

In other prototypical, one-dimensional models of Hilbert space shattering, such as the spin-1 dipole-conserving model [12, 13], or the $t$-$J_z$ model [20], it has been proven that boundary-spin autocorrelation functions decay to some nonzero, $L$-independent value in the thermodynamic limit [20, 24]. This nonzero value serves as an indirect signature of the additional conserved quantities in these models, since it cannot be accounted for using only their conventional global symmetries, and implies that charge initially localized at the boundary will remain localized there indefinitely. For the pair-flip model, however, there exists no such analytical lower bound on the decay of boundary correlation functions in the thermodynamic limit. Such a bound was computed numerically in Ref. [24] for systems of finite size (up to $L = 15$), using projectors onto Krylov sectors [as in (18)], but the behavior in the thermodynamic limit remains unclear when extrapolating from such small system sizes.

To further motivate the requirement for an analytical solution, we perform numerical simulations of the infinite-temperature boundary autocorrelation function $\langle \hat{S}_0(t)\hat{S}_0(0)\rangle_{\beta=0}$ in the translation-invariant spin-1/2 triple-flip model with Hamiltonian

$$\hat{H}_{3,m} = -J \sum_i \sum_{\alpha \neq \beta} |\alpha\alpha\alpha\rangle\langle\beta\beta\beta|_{[i,i+3]} \, . \tag{19}$$

Time evolution is performed using standard Chebyshev polynomial techniques [79], and the infinite-temperature average is performed using a stochastic approximation to the trace [79,

80]. The results are shown in Fig. 1 for systems of size $L = 16$ to 22 out to late times ($Jt = 10^3$). While the boundary autocorrelation function exhibits slow temporal decay that eventually plateaus for each $L$, it cannot be reliably inferred from the system sizes that are accessible numerically that the values of the plateaux remain nonzero in the thermodynamic limit $L \to \infty$. Indeed, at least for this limited range of system sizes, there is no appreciable convergence of the late-time plateaux with increasing $L$, which is notably in contrast with other models exhibiting fragmentation, where finite-size numerics appear to converge rapidly [12, 20].

For bulk Mazur bounds, the situation is even less clear. Reference [24] also computed numerical Mazur bounds in the pair-flip model for the spin belonging to the center of the chain, and found a convincing $1/L$ decay up to the largest numerically accessible system sizes. This result is surprising, since it suggests that the additional conserved quantities due to the conserved pattern do not qualitatively change the scaling of the Mazur bound with system size beyond the contribution from the U(1) symmetries (1). If true, this would distinguish the dynamics of the pair-flip model from, e.g., the $t$-$J_z$ model, whose conserved patterns imply a $1/\sqrt{L}$ decay of the bulk Mazur bound [20, 24]. This discrepancy between the bulk bounds in these two models, despite their seemingly similar fragmentation structure, has remained an open problem. In the next section, we prove for $p$-flip dynamics that the boundary autocorrelation functions saturate to a constant value in the thermodynamic limit, and that bulk Mazur bounds decay as $1/\sqrt{L}$ for sufficiently large $L$ (which happen to be greater than those accessible in state-of-the-art numerical simulations). We also provide a physical interpretation of this behavior and complementary numerics to corroborate the exact results.

# 4 Exact results

## 4.1 Summary of exact results

### 4.1.1 Mapping to walks on fractal structures

The first ingredient entering the Mazur bound (18), which utilizes projectors onto Krylov sectors as a set of conserved quantities, is the dimension of each Krylov sector. Since each label uniquely identifies a Krylov sector, enumerating the spin configurations that map to a particular label upon performing the decimation procedure will give the dimension of the Krylov sector that corresponds to the label. Here, we outline the procedure used to perform this enumeration exactly, with the precise details thereof presented in Sec. 4.2 onward.

For general $p$ and $m$ this task can be performed by introducing the *directed Husimi cactus* graph, depicted in Fig. 2 for $m = p = 3$. Each spin configuration in real space (in the computational basis) is in one-to-one correspondence with a *walk* on the cactus graph; moving from left to right, say, when a spin of a given color is encountered in real space, the edge of the corresponding color is traversed on the cactus. By construction, spin configurations that map to *closed* walks on the cactus graph produce the trivial label upon performing the decimation procedure. The simplest example of this is a run of length three, which maps to a closed walk around one of the 3-cycles. A less trivial example is the grouped configuration on the left-hand side of (6), whose corresponding walk is highlighted in Fig. 2. Equivalently, closed walks on the directed cactus in Fig. 2 will generate spin configurations in real space that correspond to a noncrossing configuration of grouped trimers. Furthermore, *open* walks whose initial and final positions differ by a geodesic (i.e., shortest) distance of $\ell > 0$ edges correspond to spin configurations with a nonempty label of length $\ell$. The corresponding geodesic path is precisely the Krylov sector label, and decorations of this path on the cactus graph correspond in real space to insertions of regions of noncrossing trimers. Hence, to evaluate the Krylov sector dimension $d_\mu$ corresponding to a particular irreducible label $\mu$, we must enumerate walks of

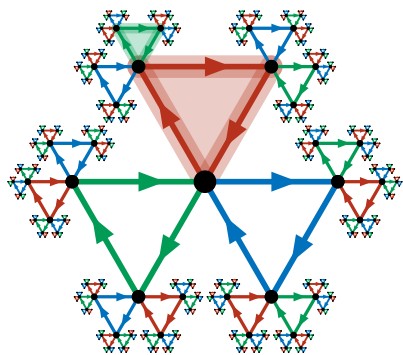

Figure 2: The directed Husimi cactus graph, used to enumerate Krylov sector dimensions for the $m = 3$ color triple-flip model. Closed walks of length $L$ correspond to computational basis states in real space described by the trivial label, while open walks on the graph map to spin configurations with a nontrivial label equal to the geodesic path between the start and end points of the walk. The walk that corresponds to the spin configuration on the left-hand side of (6) is highlighted with thick lines and the corresponding loops are shaded.

length $L$ whose endpoints are separated by a geodesic path that corresponds to the label $\mu$ on the appropriate cactus graph. Note that the generalization of Fig. 2 to generic $m$ and $p$ involves $m$ closed, directed $p$-cycles connected to every vertex. For the pair-flip model, $p = 2$, the Husimi cactus graph reduces to a Bethe lattice with coordination number $m$ (utilized in Ref. [59] in the context of deriving entanglement properties of the pair-flip model).

The second ingredient entering the bound (18) is the expectation value of the operator in question, $\hat{S}_i^z$, averaged over all states belonging to a given Krylov sector, indexed by $\mu$. Again, this can be given an interpretation in terms of enumerating constrained walks on the Husimi cactus. Making use of the indicator function $\mathbf{1}_\alpha$, we write

$$\sum_{\mathbf{s} \in \mu} \langle \mathbf{s} | \hat{S}_i^z | \mathbf{s} \rangle = \sum_{\alpha=0}^{m-1} S^z(\alpha) \sum_{\mathbf{s} \in \mu} \mathbf{1}_\alpha(S_i^z), \tag{20}$$

which shows that the left-hand side of (20) can be evaluated by enumerating all walks whose endpoints are separated by a geodesic path that corresponds to the label $\mu$ *and* whose $i$th step is constrained to be of a particular, fixed color. These counts can then be weighted by $S^z(\alpha)$ to obtain the sum over all $\mathbf{s} \in \mu$, for which the $i$th step in unconstrained. For autocorrelation functions of products of $\hat{S}_i^z$, such as $\hat{S}_i^z \hat{S}_{i+1}^z$, a decomposition analogous to (20) can be applied, in which a larger portion of the walk is fixed.

The self-similar nature of the cactus graph gives rise to closed recursion relations, which means that the task of enumerating such constrained walks can in fact be performed exactly (see Sec. 4.2, where this task is explicitly carried out). In practice, writing down a closed-form solution requires the solution to an order-$p$ polynomial; we therefore restrict our attention to the pair- and triple-flip models for simplicity. In the remainder of this subsection, we summarize the main results that follow from this exact enumeration procedure, and provide a physical interpretation thereof.

### 4.1.2 Boundary spins

The effect of the conserved patterns is strongest at the boundary of the system. Specifically, we show explicitly that the spin at either boundary of the one-dimensional system exhibits

an *infinite* autocorrelation time in the thermodynamic limit for both the pair- and triple-flip models. For the pair-flip model with $m \geq 3$,[5] we show in Sec. 4.3 that, in the thermodynamic limit, the late-time average of the edge-spin autocorrelation function satisfies

$$C_0^z \geq \frac{(m+1)(m-2)^2}{12(m-1)} = \left(\frac{m-2}{m-1}\right)^2 \langle \hat{S}_0^z(0)\hat{S}_0^z(0)\rangle_{\beta=0}. \quad (40)$$

This result, which takes into account *all* conserved quantities of the form (10), proves the conjecture in Ref. [24] that the bound saturates to an *L*-independent constant in the thermodynamic limit. The corresponding expression (43) for the triple-flip model is significantly more involved, but nevertheless remains nonzero in the thermodynamic limit for all $m \geq 2$.

The physical interpretation of this result is straightforward. As we show in Sec. 4.3, typical infinite-temperature spin configurations have labels that occupy a nonzero fraction of system size. For instance, in the pair-flip model, this fraction is $(m-2)/m$. Hence, there is a nonzero probability of finding the leftmost dot on the leftmost lattice site at infinite temperature. Next, observe the structure of the Mazur bound (18): For each initial state, one sums over all final computational basis states with the same label with equal weight. To develop nonzero auto-correlation, the leftmost dot should exhibit a nonzero return probability over the ensemble of accessible final states. Otherwise, the sum will average to zero. This behavior is seen numerically in the left panel of Fig. 3, where the probability of finding the first dot on the leftmost site at infinite temperature is shown; the scaling collapse with increasing system size shows that this probability remains nonzero in the thermodynamic limit. More precisely, we sample spin configurations at infinite temperature, perform the decimation procedure to identify the label, and record the position of the first dot in the label. While different choices of decimation procedure (e.g., right to left, or left to right) will produce $O(1)$ differences in the locations of dots,[6] the asymptotic exponential decay into the bulk is unaffected by this choice. A more rigorous version of this intuitive argument is presented in Sec. 4.1.4.

This reasoning also explains the dependence on the number of colors *m*. For large *m*, and at infinite temperature, the probability of finding a run of length *p* is suppressed. Consequently, it is to be expected that larger *m* (and larger *p*) will give rise to longer labels in typical infinite-temperature spin configurations. This implies that the first and last dots in the label will be more strongly localized with an increasing number of colors, since the dots cannot pass through one another. Indeed, taking *m* to be large, the typical label length (as a fraction of *L*) approaches 1 from below and $C_0^z$ in (40) does not decay from its initial value.

### 4.1.3 Bulk spins

The effect of the conserved pattern on bulk autocorrelation functions is more subtle. The late-time average of the autocorrelation function for spins belonging to the bulk of the system decays with increasing system size. Nevertheless, including all conserved quantities gives rise to an asymptotic $1/\sqrt{L}$ decay, which is parametrically slower than the $1/L$ decay implied by the U(1) symmetries alone (15). In Sec. 4.4, we show that, for the pair-flip model with $m \geq 3$, the late-time average of the autocorrelation function of the central spin obeys the bound

$$C_{L/2}^z \geq \sqrt{\frac{m-1}{\pi L} \frac{(m-2)(m^4-6m^3+16m^2-20m+10)}{(m-1)[m(m-2)+2]^2}} \langle \hat{S}_{L/2}^z(0)\hat{S}_{L/2}^z(0)\rangle_{\beta=0} + O\left(L^{-3/2}\right), \quad (54)$$

---

[5]Recall that the pair-flip model only exhibits an exponential-in-*L* number of Krylov sectors for $m \geq 3$. The triple-flip model, however, only requires $m \geq 2$ to exhibit Hilbert space fragmentation [see Eq. (9)].

[6]This can be proven by considering the probability that a lattice walk returns to the origin after at most *k* steps (making the location of the dot ambiguous over a lengthscale of at most *k*). For the pair-flip model, this scales asymptotically as $\sim (2\sqrt{m-1}/m)^k$. The average worst-case difference in distance therefore remains bounded.

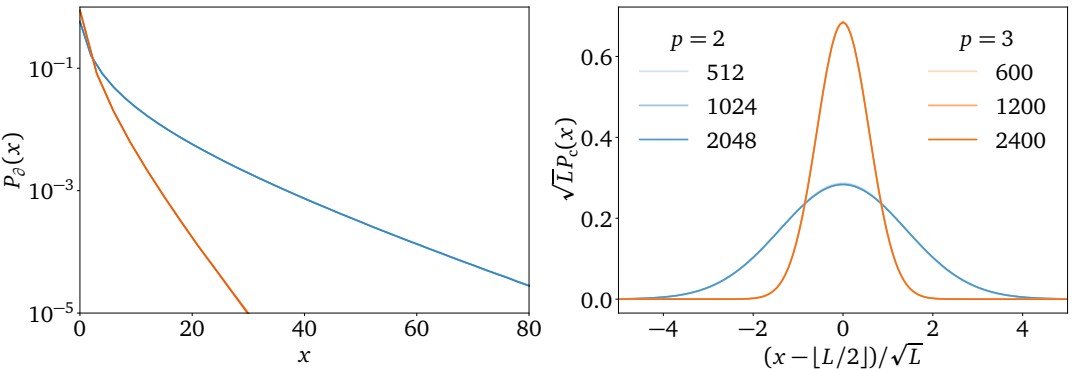

Figure 3: Left: Infinite-temperature probability distribution $P_\partial(x)$ of the leftmost dot in the label for the $m = 3$ pair- and triple-flip models for various systems sizes. The distribution decays exponentially into the bulk and is independent of system size (all curves overlap). Right: Infinite-temperature probability distribution of the central dot in the label $P_c(x)$ for the $m = 3$ pair- and triple-flip models. The distribution is centered around $L/2$ and has width $\sqrt{L}$, as verified by the scaling collapse. In both panels the positions of the dots are extracted by averaging over the label that is obtained from carrying out the decimation procedure from left to right and from right to left to obtain a symmetric distribution.

as system size $L \to \infty$ for fixed $m$. While (54) decays with $L$, it nevertheless implies anomalous thermalization: Charge initially localized in the middle of the system is able to spread out over a region of size $\sqrt{L}$, which is vanishingly small compared to the total size of the system. Furthermore, we evaluate the bound at an arbitrary position $x$ along the chain. The generalization of (54) shows that the correlation functions decay slowly with distance from the boundary as $1/\sqrt{x}$, implying that charge initially localized at position $x$ will spread out over a distance $\sqrt{x}$.

The $1/\sqrt{L}$ decay of bulk autocorrelation functions is further verified in the right panel of Fig. 3, in which we plot the probability distribution of the central dot in the label. Again, different choices of decimation procedure only give rise to an $O(1)$ difference in positions, which does not affect the scaling analysis presented in Fig. 3. We observe that the probability distribution is centered around $L/2$ and, importantly, has a width that scales as $\sqrt{L}$. Consequently, the central dot does not explore the entire system; instead, it remains *anomalously localized* near the middle of the system (at least on length scales comparable to $L$). This implies that the return probability scales as $1/\sqrt{L}$, consistent with the exact result (54). While we do not derive an analytical expression for the triple-flip model, it is clear from Fig. 3 that the central dot is – as in the pair-flip model – delocalized over a region of size $\sqrt{L}$ and the autocorrelation function of the central spin should decay similarly as $1/\sqrt{L}$.

The result (54) and its generalization to arbitrary sites (55) correct the numerical conjecture of Ref. [24] that the bulk Mazur bound decays as $1/L$. Instead, the true asymptotic $1/\sqrt{L}$ decay places the model in line with, e.g., the $t$-$J_z$ model, which exhibits similar anomalous thermalization in the bulk. Equation (54) therefore conclusively resolves the friction that previously existed between the two models: While there does exist an entropic interaction between dots in $p$-flip models (since the number of paired $p$-mers in the intervening region depends on the separation of dots) that is absent in the $t$-$J_z$ model, this feature of the dynamics *does not modify the qualitative asymptotic behavior of the bound.*

### 4.1.4 Boundary-localized conserved quantities

Since boundary autocorrelation functions remain nonzero asymptotically, there must exist sufficiently local conserved quantities at the boundaries of the system. While the conserved quantities presented in Eq. (10) are complete, they can ostensibly be generated by a more local set of operators. Inspired by Ref. [20], we deduce that the color of the $\mu$th dot in the irreducible label is also a valid conserved quantity. We will show below that the corresponding operators – referred to in Ref. [20] as statistically localized integrals of motion (SLIOMs) – are sufficient to explain the boundary localization observed in infinite-temperature dynamics (40). Consider the family of operators [20]

$$\hat{I}_\mu = \sum_{j=0}^{L-1} \hat{P}_{\mu j} \hat{S}_j^z \,, \tag{21}$$

conserved by (2), where $\hat{P}_{\mu j}$ is a projection operator, diagonal in the computational basis, that projects onto spin configurations in which the $\mu$th dot, as measured from the left, resides on site $j$. Analogous (but not orthogonal) conserved quantities can be written down for projection operators defined with respect to the right boundary. Since the ordering of dots is fixed within a Krylov sector, as is their color, the operators in Eq. (21) are all block diagonal with respect to the Hamiltonian's Krylov sectors and hence commute with (2). For the projection $\hat{P}_{\mu j}$ to be defined unambiguously, we must fix a particular choice of decimation procedure. Consider the Mazur bound that one obtains from the single conserved operator $\hat{I}_0$ corresponding to the color of the leftmost dot in the label. From (11) we have

$$\frac{(S_0^z|I_0)(I_0|S_0^z)}{(I_0|I_0)} = \frac{(\mathrm{Tr}\,\hat{P}_{0,0})^2}{D\sum_j \mathrm{Tr}\,\hat{P}_{0,j}} \langle \hat{S}_0^z(0)\hat{S}_0^z(0)\rangle_{\beta=0} \,, \tag{22}$$

if the following decimation procedure is used: from left to right, search for identically colored pairs of spins and, if such a pair is found, remove it from the system and return to the left edge. This procedure ensures that only $\hat{P}_{0,0}$ appears in the numerator of (22). It should be noted, however, that the numerical value of (22) remains unchanged if a different decimation procedure is used since $\hat{I}_0$ is invariant under the choice of decimation procedure. The expression (22) has a simple interpretation: since the fraction of states described by the empty label vanishes in the thermodynamic limit, we have $D \approx \sum_j \mathrm{Tr}\,\hat{P}_{0,j}$ and (22) is therefore related to the probability that the leftmost dot is found on the leftmost site. From Fig. 3, this probability remains finite in the thermodynamic limit,[7] so (22) immediately leads to a nonzero Mazur bound at the boundary. We have therefore identified a single conserved operator that is sufficient to establish nonzero asymptotic autocorrelation in the thermodynamic limit. Note that while $\hat{I}_0$ is not a sum of strictly local terms it is nevertheless *pseudolocalized* [61] with respect to the infinite-temperature state[8] (a generalization of the more conventional notion of *psuedolocality* [81, 82], which applies to translation-invariant operators). The existence of such boundary-localized operators for the Hamitonian (2) is reminiscent of strong zero modes [49–54], as discussed in Ref. [20].

The behavior of autocorrelation functions at the boundary of the system is therefore qualitatively similar to other fragmented models that exhibit conserved patterns, such as the one-dimensional $t$-$J_z$ model [20, 24]. This model also exhibits infinite autocorrelation times at its boundaries that may be understood in terms of its SLIOMs.

---

[7]While Fig. 3 uses a different decimation procedure from the one used to derive (22), the probability that the first dot in the label is found on the first site remains nonzero.

[8]Reference [61] actually refers to such operators as *cryptolocalized* since they cannot be written in terms of strictly local operators.

## 4.2 Enumerating walks

### 4.2.1 Closed walks

We now move to presenting the details of the calculations that underlie the main results (40), (43) and (54). We will first derive the generating function for *closed* walks on the cactus graph (Fig. 2), which is equivalent to enumerating the number of spin configurations that map to the trivial label upon performing the decimation procedure described in Sec. 2.2. Since different labels remain dynamically disconnected, this procedure therefore gives the dimension of the Krylov sector characterized by the trivial (i.e., empty) label. Denote the generating function for closed walks on the $(p, m)$ cactus ($m$ closed, directed $p$-cycles around every vertex) by $\psi_{p,m}(x)$. The number of spin configurations that map to the trivial label can then be extracted from the coefficient $[x^L]\psi_{p,m}(x)$ in a system of size $L$ spins. A generic closed walk that begins and ends at the origin (or root node) of the cactus graph can be decomposed as

 (i) the trivial walk,

 (ii) (a) hopping around one of the $m$ different $p$-cycles connected to the origin, performing a closed walk at each of the $p - 1$ vertices (that always remains at depth $\geq 1$),

     (b) performing another closed walk that begins and ends at the origin.

The "depth" of a particular vertex on the cactus graph is defined as the geodesic distance to the origin (forgetting about the directed nature of graph). Consequently, the generating function $\psi_{p,m}(x)$ decomposes as

$$\psi_{p,m}(x) = 1 + mx^p R_{p,m}^{p-1}(x)\psi_{p,m}(x), \tag{23}$$

which is written in terms of the generating function $R_{p,m}(x)$ for closed walks that begin and end at depth[9] $d$, always remaining at a depth greater than or equal to $d$. The function $R_{p,m}(x)$ satisfies an analogous decomposition

$$R_{p,m}(x) = 1 + (m-1)x^p R_{p,m}^p(x). \tag{24}$$

The factor $m-1$ appears in (24) since there are only $m-1$ edges connected to any vertex that is not the origin that do not lower the depth. Hence, to find $R_{p,m}(x)$, and therefore $\psi_{p,m}(x)$, we are required to solve an order-$p$ polynomial (24), choosing the root that satisfies $R_{p,m}(x) \to 1$ as $x \to 0$. We solve (23) and (24) exactly for the pair- and triple-flip models for a generic number of colors $m$. For the pair-flip model, the solutions are

$$R_{2,m}(x) = \frac{1 - \sqrt{1 - 4(m-1)x^2}}{2(m-1)x^2}, \qquad \psi_{2,m}(x) = \frac{2(m-1)}{m - 2 + m\sqrt{1 - 4(m-1)x^2}}, \tag{25}$$

consistent with, e.g., Refs. [59, 83]. For the triple-flip model, the solutions derive from a cubic equation and are therefore significantly more involved. Nevertheless, they can be written in closed form using trigonometric functions:

$$R_{3,m}(x) = \sqrt{\frac{4}{3(m-1)x^3}} \sin\left[\frac{1}{3}\arctan\left(\sqrt{\frac{27(m-1)x^3}{4 - 27(m-1)x^3}}\right)\right], \tag{26a}$$

$$\psi_{3,m}(x) = \frac{3(m-1)}{m - 3 + 2m\cos\left[\frac{2}{3}\arctan\left(\sqrt{\frac{27(m-1)x^3}{4 - 27(m-1)x^3}}\right)\right]}. \tag{26b}$$

As required, in both of the above expressions, $[x^L]\psi_{p,m}(x)$ is only nonzero when $L$ is a multiple of $p$, since trivial labels must be composed of an integer number of $p$-mers.

---

[9]Note that the self-similarity of the Husimi cactus implies that this generating function is independent of depth.

#### 4.2.2 Open walks

Open walks on the directed cactus graph can be enumerated in a similar fashion. To evaluate the number of states in a Krylov sector with a label of length $\ell$, we are required to count the number of walks from the origin to some other site $s$, which is separated from the origin by a geodesic distance of $\ell$ edges on the directed graph. Define the set of edges and vertices on the cactus graph belonging to the geodesic path as $\{e_j\}_{j=1}^{\ell}$ and $\{v_k\}_{k=0}^{\ell}$, respectively. The walk from the origin ($v_0$) to the vertex $v_\ell$ at the end of the label can be decomposed as follows:

(i) (a) hopping along edge $e_1$ corresponding to the first color in the label,

    (b) performing a walk from vertex $v_1$ to vertex $v_\ell$,

(ii) (a) performing a *closed* walk, beginning on one of the $m-1$ edges that are complementary to $e_1$,

    (b) performing a walk from the origin $v_0$ to vertex $v_\ell$.

The walk from vertex $v_1$ to vertex $v_\ell$ can be decomposed analogously. Hence, denoting the generating function for open walks between sites separated by $\ell$ bonds on the $(p, m)$ cactus by $T_{p,m}^{(\ell)}(x)$, we have the decomposition

$$T_{p,m}^{(\ell)}(x) = x\,T_{p,m}^{(\ell-1)}(x) + (m-1)x^p R_{p,m}^{p-1} T_{p,m}^{(\ell)}(x). \tag{27}$$

This recursion relation is solved by

$$T_{p,m}^{(\ell)}(x) = [xR_{p,m}(x)]^\ell \psi_{p,m}(x) \equiv f_{p,m}^\ell(x)\psi_{p,m}(x), \tag{28}$$

where $R_{p,m}(x)$ and $\psi_{p,m}(x)$ are given by (25) and (26) for the pair- and triple-flip models, respectively, and we have introduced $f_{p,m}(x) = xR_{p,m}(x)$. Importantly, the number of walks that correspond to a particular label depends only on the length $\ell$ of the label, not on its individual structure. As required, the nonzero coefficients $[x^n]T_{p,m}^{(\ell)}(x)$ satisfy $(n - \ell) \in p\mathbb{N}$ since a system must have at least $L = \ell$ sites to host a label of length $\ell$, and decorations of the walk with noncrossing $p$-mers can only increase the length of the walk by integer multiples of $p$. As a further sanity check, the Krylov sector dimensions summed over all labels must be equal to the total size of the Hilbert space $m^L$. This is verified by composing the various generating functions, which satisfy

$$\psi_{p,m}(x)G_{p,m}(f_{p,m}(x)) = (1 - mx)^{-1} = \sum_{n=0}^{\infty} m^n x^n, \tag{29}$$

where $G_{p,m}(x)$ from (8) counts the number of labels of a particular length. Since the $[x^L]$ coefficient of (29) is $m^L$, the Krylov sector dimensions indeed sum to the total Hilbert space dimension for $m$-level degrees of freedom.

### 4.3 At the boundary

#### 4.3.1 Finite system size

Given the generating functions derived in Sec. 4.2, it is straightforward to write down an exact expression for the spin autocorrelation function at the boundary. In this case, the summation over spin configurations $\mathbf{s}$ belonging to Krylov sector $\mu$ depends only on the color of the first dot in the irreducible label:

$$\sum_{\mathbf{s}\in\mu} \langle \mathbf{s}|\hat{S}_e^z|\mathbf{s}\rangle \to \left(\frac{m-1-2\alpha}{2}\right)\left[T_{m,p}^{(\ell-1)}(x) - T_{m,p}^{(\ell+p-1)}(x)\right], \quad \text{if first dot } \alpha, \tag{30}$$

where the $[x^{L-1}]$ coefficient of the right-hand side equals the summation over spin configurations in a system of size $L$ spins. As noted in Sec. 4.1, this is because only the first dot in the label is able to visit the leftmost site of the system. Suppose, for instance, that the first dot in the label is red. If the first step of the walk is *also* red, then the walk need only traverse the remaining $\ell-1$ colors in the irreducible label in the $L-1$ remaining steps. If, on the other hand, the first step is *not* red, the walk must instead traverse $\ell+p-1$ colors in the remaining $L-1$ steps. Hence,

$$C_0^z \ge \frac{1}{12}(m^2-1)\sum_{\ell>0} g_\ell \frac{s_\ell^2(L)}{d_\ell(L)D(L)}, \quad \text{where} \quad s_\ell(L) = [x^{L-1}]\left\{T_{m,p}^{(\ell-1)}(x)-T_{m,p}^{(\ell+p-1)}(x)\right\}, \quad (31)$$

where all dependence on $p$ and $m$ is left implicit, and the coefficient $g_\ell$ denotes $[x^\ell]G_{p,m}(x)$ (counting the number of labels of length $\ell$). Since the Krylov sector dimension depends only on system size and the length of the label through (28), we have written $d_\mu \to d_\ell(L)$ for a system of size $L$ sites.

### 4.3.2 Asymptotic expressions

The solution (31) is exact, and may be used to bound $C_0^z$ for large but finite system sizes by extracting the relevant coefficients from the generating functions derived in Sec. 4.2. However, the real power of the expression lies in the fact that we are able to utilize asymptotically exact (saddle-point) expressions for these coefficients to extract the behavior of (31) in the thermodynamic limit $L \to \infty$.

We require an asymptotic expansion of the coefficients $d_\ell(L)$ and $s_\ell(L)$ appearing in (31) valid for $\ell, L \to \infty$ with $\ell/L$ fixed, i.e., for $L = \Theta(\ell)$. For $f(z)$ and $\psi(z)$ satisfying certain technical conditions [84], we have

$$[z^L]\{f^\ell(z)\psi(z)\} = \frac{f(\rho)^\ell \psi(\rho)}{\rho^{n+1}\sqrt{2\pi\ell\,\delta f(\rho)}}(1+o(1)), \quad (32)$$

where $\rho$ is a positive, real solution to the saddle-point equation $\Delta f(\rho) = L/\ell = y^{-1}$, smaller than the radii of convergence of $f(z)$ and $\psi(z)$. This equation defines the saddle-point radius $\rho(y)$ for each label length $\ell$, parameterized by the continuous variable $y = \ell/L$. Note, however, that some lattice-level detail remains: (32) must vanish for all $\ell$ that satisfy $\ell \notin \{L-pn\}$. Correspondingly, for the generating function (28), there are in fact $p$ solutions occurring at the same radius in the complex plane, leading to an additional factor of $p$. The two derivative operators $\Delta$ and $\delta$ are defined as

$$\Delta f(z) = z\frac{f'(z)}{f(z)}, \qquad \delta f(z) = \frac{1}{z}(\Delta f)'(z). \quad (33)$$

We can alternatively write the expression (32) for the nonzero coefficients, including the contribution from the $p$ saddle points, as

$$[z^L]\{f^\ell(z)\psi(z)\} = \frac{p}{\sqrt{L}}C(y)\exp[L\Phi(y)](1+o(1)), \quad (34)$$

where we have defined the $L$-independent functions

$$C(y) = \frac{\psi(\rho)}{\rho\sqrt{2\pi y\,\delta f(\rho)}}, \quad (35a)$$

$$\Phi(y) = y\ln f(\rho) - \ln\rho, \quad (35b)$$

where $\rho$ is considered a function of $y$ via the saddle-point equation. The equations (34) and (35) already provide an asymptotic approximation to $d_\ell(L)$, accurate up to $o(1)$ multiplicative corrections. Using the saddle-point equation, and the definitions of the derivative operators in (33), we find that $\Phi(y)$ satisfies

$$\Phi'(y) = \ln f(\rho), \tag{36a}$$

$$\Phi''(y) = -[\rho^2 y^3 \delta f(\rho)]^{-1}. \tag{36b}$$

The coefficient $s_\ell(L)$ (31) requires some additional effort to produce a convenient asymptotic expression. We must evaluate the expression (34) at the values $y = [\ell + (p-2)/2 \pm p/2]/(L-1)$. Since $p$ is assumed to be $O(1)$, we can expand in gradients of $\Phi(y)$:

$$s_\ell(L) = \frac{p}{\sqrt{L-1}} C(y) e^{(L-1)\Phi(y) + \Phi'(y)\left(y + \frac{p-2}{2}\right)} \left[ e^{-\frac{p}{2}\Phi'(y)} - e^{\frac{p}{2}\Phi'(y)} \right] (1 + o(1)). \tag{37}$$

Hence, for $\ell \in \{L - pn\}$ the combination of $s_\ell(L)$ and $d_\ell(L)$ that appears in the bound for $C_0^z$ has the asymptotic expression

$$\frac{s_\ell^2(L)}{d_\ell(L)} = \frac{p\sqrt{L}}{L-1} C(y) e^{(L-2)\Phi(y) + \Phi'(y)(2y + p - 2)} \left[ e^{-\frac{p}{2}\Phi'(y)} - e^{\frac{p}{2}\Phi'(y)} \right]^2 (1 + o(1)). \tag{38}$$

To complete the bound for $C_0^z$ (31), we must perform the remaining summation over valid label lengths $\ell$. This can be achieved by representing the summation as an integral and performing a saddle-point approximation thereof in the thermodynamic limit. We will perform this task explicitly for both the pair- and triple-flip models.

**Pair-flip model.** In the pair-flip model, the number of labels of length $\ell$ grows as $g_\ell = m(m-1)^{\ell-1}$. Representing the summation over $\ell$ in (31) as an integral over the variable $0 \le y \le 1$, the saddle-point equation for $y = \ell/L$ is therefore

$$\ln(m-1) + \Phi'(y) \overset{!}{=} 0, \tag{39}$$

which is solved by $y_\star = (m-2)/m$, or, equivalently, $\rho_\star = 1/m$. Note that this is the same saddle-point equation as one would have obtained if evaluating Eq. (29) using a saddle-point approximation. We can use this to deduce that $\rho_\star = 1/m$ is in fact the solution for *generic* $p$. Furthermore, we may interpret the solution $y_\star$ as the most probable label length that dominates the summation over $\ell$. Equivalently, if a state is drawn at random from the computational basis at infinite temperature, it will belong to a Krylov sector with $\ell/L = (m-2)/m$ in the thermodynamic limit.[10] We note in passing that, since $(m-2)/m < 1$, this result proves that pair-flip models exhibit strong shattering (as defined in Refs. [12, 13]) for all $m \ge 3$, i.e., in the thermodynamic limit, the largest sector (which occurs for $\ell = 0$) will be selected with probability zero.

Using the saddle-point method to evaluate the integral over $y$ as $L \to \infty$, we arrive at the final expression

$$C_0^z \ge \frac{(m+1)(m-2)^2}{12(m-1)} = \left(\frac{m-2}{m-1}\right)^2 \langle \hat{S}_0^z(0)\hat{S}_0^z(0)\rangle_{\beta=0}, \tag{40}$$

with finite-size corrections of magnitude $O(L^{-1})$. That is, the time-averaged boundary autocorrelation function saturates to a nonzero, $L$-independent constant in the thermodynamic limit for $m \ge 3$ colors. For $m = 3$ colors, the expression (40) evaluates to $1/6$, proving the convergence of the numerics in Ref. [24].

---

[10]The distribution is peaked at $\ell/L = (m-2)/m$ with a width that scales as $\sim L^{1/2}$. This explains the dominance of the $L/3$ label length for the pair-flip model observed in Ref. [47].

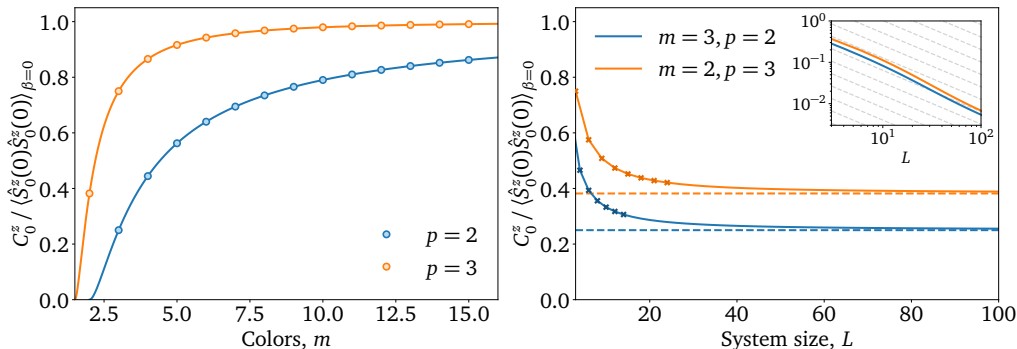

Figure 4: Left: Mazur bound for the time-averaged autocorrelation function for the boundary spins in the pair- (blue) and triple-flip (orange) models in the thermodynamic limit. Right: How the $L \to \infty$ limit is approached in systems of finite size for $m = 3$ colors [i.e., spin-1] in the pair-flip model (blue), and $m = 2$ colors [i.e., spin-1/2] in the triple-flip model (orange). The difference between the solid and dashed lines is shown in the inset, which illustrates the slow $1/L$ convergence (gray dashed lines) of the result to its asymptotic value. Exact numerical enumeration of Eq. (18) is shown using crosses to validate the analytical results.

**Triple-flip model.** The growth of the number of labels in the triple-flip model for generic $m$ generalizes the Fibonacci numbers in Eq. (9). The exact growth is

$$g_\ell = \frac{m}{(m-1)\sqrt{(m+3)(m-1)}} \left( \frac{m-1+\sqrt{(m+3)(m-1)}}{2} \right)^{Ly+1} (1 + o(1)).  \tag{41}$$

As noted below Eq. (39), the solution to the saddle-point equation corresponding to the integral over $y$ is still $\rho_\star = 1/m$. However, inverting this expression to find the most probable label length is more involved. Nevertheless, a closed-form expression can still be written down:

$$y_\star(m) = \frac{4m-6}{3 - 2m - 3\sqrt{\frac{3(m-1)}{m+3}} \cot\left( \frac{1}{3} \arctan\left( \frac{3}{3-2m} \sqrt{\frac{3(m-1)}{m+3}} \right) \right)},  \tag{42}$$

which monotonically approaches unity from below for $m \geq 2$. The bound for the boundary autocorrelation function follows from applying the saddle-point method to the integral over $y$ as $L \to \infty$. The final expression for the triple-flip model is

$$C_0^z \geq \frac{\left( m-1+\sqrt{(m+3)(m-1)} \right)}{2m(m-1)\sqrt{(m+3)(m-1)}} \left( \frac{f(\rho_\star)^3 - 1}{f(\rho_\star)} \right)^2 \psi(\rho_\star) y_\star \langle \hat{S}_0^z(0) \hat{S}_0^z(0) \rangle_{\beta=0},  \tag{43}$$

with finite-size corrections of magnitude $O(L^{-1})$, where $f(x)$ and $\psi(x)$ are defined in (28). The expression that multiplies the initial value of the correlation function, $\langle \hat{S}_0^z(0) \hat{S}_0^z(0) \rangle_{\beta=0}$, is strictly positive and less than unity for all $m \geq 2$, proving that the time-averaged boundary autocorrelation function again saturates to a nonzero, $L$-independent constant. The behavior of the expression (43) as a function of $m$, and the convergence of the exact result (31) to the asymptotic expression (43) is illustrated in Fig. 4.

## 4.4 In the bulk

Having shown that the late-time average of boundary autocorrelation functions remains finite in the thermodynamic limit, we now move to quantifying the late-time behavior of autocorrelation functions in the *bulk* of the system. The calculation is complicated significantly by

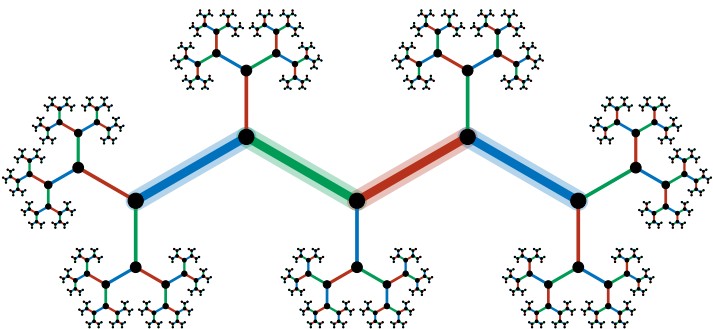

Figure 5: A convenient way to redraw the Bethe lattice for a label of length $\ell = 4$. Spin configurations of length $L$ in real space map to walks of length $L$ steps on the Bethe lattice. The "backbone," which corresponds to the irreducible label, has been highlighted using thick lines. Spin configurations that produce the highlighted irreducible label under the decimation procedure described in Sec. 2.2 must traverse the entire backbone during the walk.

the fact that the sum over spin configurations belonging to a particular Krylov sector depends on specific details of the label. Nevertheless, since the bound depends only on the sum over *all* such labels, we show that we only require knowledge of the correlation functions of dots averaged over valid irreducible labels at infinite temperature, which can be evaluated exactly.

### 4.4.1 Finite system size

From Sec. 3, using the projectors onto Krylov sectors as mutually orthogonal conserved quantities gives rise to the Mazur bound

$$C_i^z \geq \frac{1}{D} \sum_\mu \frac{1}{d_\mu} \left[ \sum_{\mathbf{s} \in \mu} \langle \mathbf{s} | \hat{S}_i^z | \mathbf{s} \rangle \right]^2 , \tag{18}$$

where $\mu$ runs over all Krylov sectors (dimensions $d_\mu$), which correspond to distinct irreducible labels. We will first be consider the case in which $i$ belongs to the center of the chain. The generalization to sites displaced from the center of the chain will follow straightforwardly. For simplicity, we focus on chains with $L$ odd, such that the problem is symmetric about the central site, and we will restrict our attention to the pair-flip model ($p = 2$). While the exact solution for the triple-flip model will be quantitatively different, the asymptotic scaling with system size ($1/\sqrt{L}$) should remain unaltered. To evaluate (18), we are therefore tasked with enumerating walks such that the color of the $(n+1)$th step is fixed, where $L = 2n+1$.

In contrast to the bound for the boundary spin (Sec. 4.3), which depends only on the first color in the irreducible label [see, e.g., Eq. (30)], enumerating walks in which the $(n+1)$th step is fixed depends on the entire content of the label (at least for labels of length $\ell \leq n$). This follows from the fact that only the first dot in the label may touch the boundary site, while the central site can, for sufficiently short labels, encounter *every* dot belonging to the label. Consider a Krylov sector $\mu$, corresponding to a label of length $\ell$ that consists of dots taking the

values $\{S_a(\mu)\}_{a=0}^{\ell-1}$. We find that the summation over spin states belonging to $\mu$ can be written

$$\sum_{\mathbf{s}\in\mu}\langle\mathbf{s}|\hat{S}_i^z|\mathbf{s}\rangle = \sum_{a=0}^{\ell}S_a(\mu)\underbrace{\{d_a(n)d_{\ell-a-1}(n)+d_{a+1}(n)d_{\ell-a}(n)\}}_{A_{a,\ell}(n)}$$

$$+\sum_{a=0}^{\ell}[S_a(\mu)+S_{a-1}(\mu)]\underbrace{\sum_{k=0}^{\infty}(-1)^{k+1}\{d_{a+k}(n)d_{\ell-a+k+1}(n)+d_{a+k+1}(n)d_{\ell-a+k}(n)\}}_{B_{a,\ell}(n)},$$

$$(44)$$

where, for convenience, we define $S_a(\mu)\equiv 0$ for $a=\ell,-1$ at the boundaries of the system. Recall that the integers $d_\ell(n)$ are given by the coefficient of $[x^n]$ in the generating function (28), which enumerates open walks of length $n$ whose endpoints are separated by $\ell$ edges. Consider drawing the Bethe lattice as in Fig. 5, where the label forms a "backbone" that must be traversed during the walk, and excursions from the backbone may take place on treelike "offshoots" connected to each vertex of the backbone. The terms on the top line correspond to walks where the $(n+1)$th step lives on the backbone, while walks in which the $(n+1)$th step occurs on one of the offshoots contribute to the second line. The factor $(-1)^{k+1}$ comes from counting the number of edges of each color at depth $k$ in an offshoot: Take an offshoot whose first edge has color $\alpha$ and denote the number of edges between vertices at depths $k$ and $k+1$ with color $\alpha$ (complementary to $\alpha$, respectively) by $N_k^{(\alpha)}$ ($\bar{N}_k^{(\alpha)}$, respectively). Then,

$$N_k^{(\alpha)}=\frac{1}{m}\left[(m-1)^k+(m-1)(-1)^k\right], \tag{45a}$$

$$\bar{N}_k^{(\alpha)}=\frac{1}{m}\left[(m-1)^k-(-1)^k\right]. \tag{45b}$$

Given two adjacent colors $\alpha_1$, $\alpha_2$ belonging to the backbone, the offshoots that emanate from the vertex between the two colors satisfy

$$\sum_{\alpha\in\{\alpha_1,\alpha_2\}}S(\alpha)\bar{N}_k^{(\alpha)}+\sum_{\alpha\notin\{\alpha_1,\alpha_2\}}S(\alpha)N_k^{(\alpha)}=[S(\alpha_1)+S(\alpha_2)](-1)^{k+1}, \tag{46}$$

due to an exact cancellation between the first terms in (45), where $S(\alpha)=(m-1-2\alpha)/2$ parameterizes eigenvalues of $\hat{S}^z$ in terms of the colors $\alpha$. To evaluate (44), we can decompose the sum over Krylov sectors in (18) into a sum over irreducible label lengths $\ell$ and a sum over irreducible labels of fixed length (denoted $\mu\in\ell$):

$$C_i^z\geq\frac{1}{D}\sum_\ell\frac{g_\ell}{d_\ell}\left(\frac{1}{g_\ell}\sum_{\mu\in\ell}\left[\sum_{\mathbf{s}\in\mu}\langle\mathbf{s}|\hat{S}_i^z|\mathbf{s}\rangle\right]^2\right), \tag{47}$$

where $g_\ell=m(m-1)^{\ell-1}$ for $\ell>0$ in the pair-flip model. This rewriting allows us to interpret the term in the parentheses as an infinite-temperature average over valid irreducible labels of fixed length $\ell$, written $\langle\cdots\rangle_{\mu\in\ell}$. The only terms in the square of (44) that depend on the details of the label are terms of the form $S_a(\mu)S_b(\mu)$. The infinite-temperature average of such terms over valid labels $\mu\in\ell$ can be evaluated exactly. The average is made nontrivial by the constraint that the label is irreducible, i.e., it contains no neighboring identical colors. Given a spin of some fixed color in the label, valid configurations can be counted by introducing the transfer matrix

$$\mathcal{T}_{\alpha\beta}=1-\delta_{\alpha\beta}\implies(\mathcal{T}^n)_{\alpha\beta}=\frac{1}{m}\left[(m-1)^n-(-1)^n\right]+(-1)^n\delta_{\alpha\beta}. \tag{48}$$

The vanishing diagonal matrix elements of $\mathcal{T}$ enforce the constraint of no repetition, such that, given spins with colors $\alpha$ and $\beta$ separated by $n$ edges, the number of valid nonrepeating spin configurations in the intervening region is given by $(\mathcal{T}^n)_{\alpha\beta}$. We therefore obtain

$$\langle S_a(\mu) S_b(\mu)\rangle_{\mu\in\ell} = \frac{(m-1)^{L-1-n}}{m(m-1)^{L-1}} \sum_{\alpha\beta} S(\alpha)(\mathcal{T}^n)_{\alpha\beta} S(\beta) = \frac{1}{12}(m^2-1)\left(\frac{-1}{m-1}\right)^n. \tag{49}$$

The correlation function $C_{ab}(\ell) \equiv \langle S_a(\mu) S_b(\mu)\rangle_{\mu\in\ell}$ therefore decays exponentially with distance with an alternating sign that depends on the parity of the separation between the two sites. This result is to be contrasted with the behavior of the infinite-temperature average of two spins (not restricted to spin configurations that form irreducible labels): $\langle \hat{S}_i^z \hat{S}_j^z \rangle_{\beta=0} = \delta_{ij}(m^2-1)/12$. Summing over the restricted set of spin configurations that form irreducible labels therefore induces short-ranged correlations between the spins that form the label; for two dots separated by a large distance, the constraint has almost no effect, whereas two neighboring spins must clearly be anticorrelated since their colors cannot match. Combining all of the above results, we arrive at an exact expression for $C_{L/2}^z$ in systems of finite size:

$$\frac{1}{12}\frac{m(m+1)}{m^L}\sum_\ell \frac{(m-1)^\ell}{d_\ell}\sum_{a,b=0}^\ell \Big\{ C_{ab}A_{a,\ell}(n)A_{b,\ell}(n) + 2[C_{ab}+C_{a,b-1}]A_{a,\ell}(n)B_{b,\ell}(n)$$
$$+ [C_{ab}+C_{a,b-1}+C_{a-1,b}+C_{a-1,b-1}]B_{a,\ell}(n)B_{b,\ell}(n)\Big\}, \tag{50}$$

with $A_{a,\ell}(n)$ and $B_{a,\ell}(n)$ defined in Eq. (44). Performing the average over labels of fixed length exactly has allowed us to reduce the summation over exponentially many Krylov sectors in (18) to an expression with a number of terms that scales only polynomially in system size $L$. Hence, Eq. (50) can already be utilized to extract the bound $C_{L/2}^z$ numerically for large (but finite) system sizes. The result of evaluating (50) numerically is shown in Fig. 6, which illustrates that the initial $\approx 1/L$ decay eventually gives way to a slower $1/\sqrt{L}$ decay at large system sizes $L \gtrsim 40$. Note that this crossover approximately coincides with the lengthscale over which dots are localized in Fig. 3, which can be shown to be $\log[m/(2\sqrt{m-1})]^{-1} \approx 40$ for the pair-flip model. However, once again, the real power of the expression (50) is that we may use asymptotic expressions for the generating function coefficients, derived in Sec. 4.3.2, with which we can extract the exact behavior of (50) in the thermodynamic limit.

### 4.4.2 Asymptotic expressions

Evaluating Eq. (50) for $L \to \infty$ requires similar techniques to those utilized in Sec. 4.3.2. Namely, we make use of asymptotic expressions for the coefficients $d_\ell(L)$, valid for $\ell, L \to \infty$ with $L = \Theta(\ell)$. The summations over positions in the label and over label lengths can then be performed using two sequential saddle-point approximations.

We will first obtain the asymptotic scaling with $L$ by neglecting all $O(1)$ multiplicative constants. To evaluate asymptotic expressions for the generating function coefficients, we introduce the two variables $x = a/n$ and $y = \ell/2n$ [note that this definition, which is more convenient in the present context, differs slightly from the variable $y$ defined below Eq. (32)]. Up to multiplicative, $L$-independent corrections, the asymptotic behavior of the integers $A_{a,\ell}(n)$ and $B_{a,\ell}(n)$ as $L \to \infty$ is

$$A_{a,\ell}(n) \sim B_{a,\ell}(n) \sim \frac{4}{n} C(x) C(2y-x) e^{n\Phi(x)+n\Phi(2y-x)}, \tag{51}$$

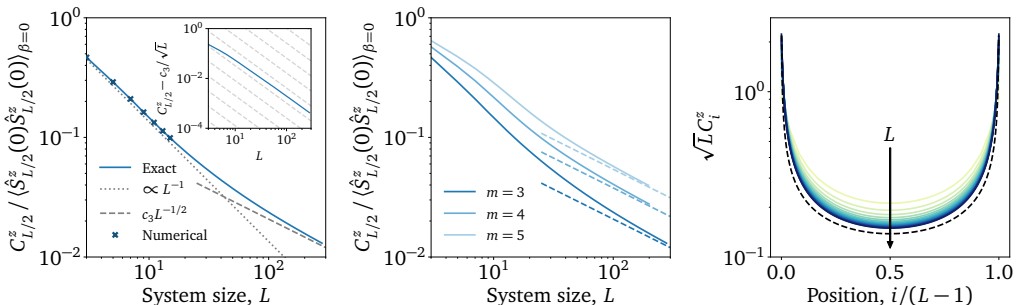

Figure 6: Left: The exact expression (50) for the bulk Mazur bound for the $m = 3$ pair-flip model evaluated at the central site of a chain with $L$ odd (blue solid line). An initial $\sim 1/L$ decay gives way to asymptotic $1/\sqrt{L}$ decay at $L \sim 40$. The analytical asymptotic expression is plotted using a gray dashed line. The inset shows the convergence of the finite-size result (50) to the large-$L$ result (54) (the light gray dashed lines have gradient $-3/2$, which is the result expected from subleading corrections to the saddle-point method). Middle: Behavior with an increasing number of colors. For all $m$ the asymptotic behavior is $\propto 1/\sqrt{L}$ with the prefactor $c_m$ defined by Eq. (54). Right: Dependence of the bound on position. Various $L$ are plotted from $L \in [25, 175]$ in equally space intervals, evaluated using the finite-size result (50), showing convergence towards the large-$L$ result (55).

where the functions $C(x)$ and $\Phi(x)$ are defined in Eq. (35). The summation over[11] $a$ in (50) can be converted into an integral over the continuous variable $x$ between $0$ and $2y$ via the replacement $\sum_{a=0}^{\ell} \to n \int_0^{2y} dx$, which, as a result of (51), is dominated by the saddle point $x_\star = y$ for all label lengths $y$. The residual Gaussian integral from this first saddle-point approximation is found by expanding the function in the exponent, which contributes a factor $\sqrt{2\pi/[4n\Phi''(y)]}$. The summation over $\ell$ can similarly be evaluated by converting to a integral over the continuous variable $y$ and using a second saddle-point approximation. The saddle-point for the integral over $y$ is determined by

$$\ln(m-1) + \Phi'(y) \overset{!}{=} 0 \,, \tag{52}$$

and the residual Gaussian integral contributes $\sqrt{2\pi/[2n\Phi''(y_\star)]}$. Note that, as expected, the saddle-point solution to (52), $y_\star = (m-2)/m$, is identical to that obtained from (39) in order to cancel the exponential growth of the Hilbert space dimension in the bound, $D(L) = m^L$. Combining these results and neglecting all $O(1)$ factors, we arrive at

$$C_{L/2}^z \gtrsim \frac{1}{n} \left( \sqrt{\frac{2\pi}{n\Phi''(y_\star)}} \right)^2 L\sqrt{L} \sim \frac{1}{\sqrt{L}} \,, \tag{53}$$

since $n = (L-1)/2$, which clearly illustrates the asymptotic $1/\sqrt{L}$ decay of the bulk autocorrelation function. If all $O(1)$ prefactors are accounted for, we can obtain the exact $m$ dependence of the bound. The final result, for all $m > 2$, is

$$C_{L/2}^z \geq \sqrt{\frac{m-1}{\pi L}} \frac{(m-2)(m^4 - 6m^3 + 16m^2 - 20m + 10)}{(m-1)[m(m-2)+2]^2} \langle \hat{S}_{L/2}^z(0)\hat{S}_{L/2}^z(0) \rangle_{\beta=0} + O(L^{-3/2}) \,. \tag{54}$$

Deriving this result requires some care since the precise asymptotic behavior of the functions $A_{a,\ell}(n)$ and $B_{a,\ell}(n)$ depends on the parity of the integer $a$ [recall that $d_a(n)$ is only nonzero if

---

[11]The summation over $b$ has a contribution from an $O(1)$ region in the vicinity of $b = a$ due to the exponential decay of the correlation matrix $C_{ab}(\ell)$ with distance $|a - b|$.

*a* and *n* have the same parity]. Nevertheless, a continuum approximation can still be written down by separating the summations over *a* and *b* in (50) into their even and odd contributions.

### 4.4.3 Interpolating between bulk and boundary

Finally, we can calculate how the bound (50) is modified if it is evaluated for a spin that is displaced from the center of the system. In particular, suppose that $L = n_L + n_R + 1$ and that we are interested in bounding the autocorrelation function of the $(n_L + 1)$th spin. Note that $n_L$ and $n_R$ are assumed to scale with system size as $n_L, n_R = \Theta(L)$. Hence, the result is not expected to reproduce (40), in which $n_L = O(1)$. For $n_L$ and $n_R$ that diverge with system size, the only modification of (54) is the replacement

$$\frac{1}{\sqrt{L}} \to \frac{1}{2}\sqrt{\frac{L}{n_L n_R}}. \tag{55}$$

Consequently, we observe that the bound decays as $1/\sqrt{x}$ with distance $x$ from the boundary of the system. As required, the full expression (55) is symmetric under exchange of $n_L \leftrightarrow n_R$.

## 5 Conclusion

We have introduced a family of models that exhibit *p*-flip dynamics, in which spins can only flip in contiguous regions of size *p* if all spins in the region are in the same state. This dynamics conserves a nonlocally encoded pattern of spins that is obtained via a real-space decimation procedure; all states that produce the same output upon performing the decimation procedure belong to the same dynamically disconnected (i.e., Krylov) sector. The pair-flip model ($p = 2$) exhibits Hilbert space fragmentation only for qutrits and higher, while models with $p > 2$ also exhibit fragmentation in systems composed of qubits. Hence, the triple-flip model introduced herein may be of independent interest, being the smallest *p* for which fragmentation occurs in spin-1/2 systems. For general *p*-flip dynamics, we showed how infinite-temperature autocorrelation functions can be bounded via an exact mapping to a problem in combinatorics: enumerating (constrained) walks on fractal structures (specifically, on directed Husimi cactus graphs). By solving this problem explicitly for $p = 2$ (pair-flip) and $p = 3$ (triple-flip) dynamics, we are able to prove various results that inform us about the thermalization properties of models exhibiting *p*-flip dynamics.

First, we proved that the conserved patterns give rise to infinite autocorrelation times for spins at the boundary of the system. Hence, for charge initially localized at the boundary, some fraction will remain there indefinitely, even in the thermodynamic limit. Second, we proved that the late-time average of bulk autocorrelation functions decays with system size $L$ as $1/\sqrt{L}$ for sufficiently large systems. This result is in sharp contrast with the $1/L$ decay that has been predicted numerically [24, 60], and shows that the conserved pattern implied by *p*-flip dynamics does indeed have an important impact on bulk thermalization properties. Namely, that charge initially localized in the middle of the system remains anomalously localized within a region of size $\sqrt{L}$ (similar to the zero-energy state in 1D systems with pure off-diagonal disorder [85]). Prior to this work, there existed tension between various canonical models of Hilbert space fragmentation: models with seemingly similar conserved patterns appeared to exhibit disparate thermalization properties in the bulk. This paper conclusively settles the debate by providing an exact solution, placing the behavior of models that exhibit *p*-flip dynamics in line with the $t$-$J_z$ model [20, 86], which provably exhibits qualitatively similar anomalous localization in the bulk [20, 24].

One direction for future work relates to higher dimensions. The naïve generalization of the pair-flip model to two dimensions (2D) has exponentially many frozen states, but does not appear to exhibit any larger Krylov sectors [24]. On the other hand, the models of topologically robust fragmentation in Ref. [47], in which the U(1) symmetries of the pair-flip model are upgraded to higher-form U(1) symmetries, exhibit fragmentation structure analogous to the 1D model, with the "dots" appearing in the irreducible label being replaced by hypersurfaces that satisfy the same criteria. While the problem of enumerating states in these higher-dimensional models is significantly more challenging, it is conceivable that the methods presented herein could be utilized to understand the behavior of autocorrelation functions when the conserved pattern consists of these extended objects.

The fragmentation exhibited by fully generic $p$-flip models is classical in the sense of Ref. [24], since fragmentation occurs in a product-state basis. However, for special choices of parameters entering the pair-flip Hamiltonian, the model reduces to a Temperley-Lieb model [87,88], which exhibits a larger local SU($m$) symmetry, with $m$ the local Hilbert space dimension. Such models exhibit fragmentation in an intrinsically entangled basis, which implies that Eq. (18) no longer represents a tight bound. Being a special case of the pair-flip Hamiltonian, Temperley-Lieb models exhibit at least as many conserved quantities as the pair-flip model, and therefore the Mazur bounds presented in this paper can nevertheless be utilized to bound the late-time behavior of autocorrelation functions. However, do there exists refinements of the exact enumeration procedure that are able to capture the more subtle fragmentation structure of Temperley-Lieb models to obtain tighter bounds?

Furthermore, we have focused exclusively on *infinite-temperature* autocorrelation functions, which average indiscriminately over all states. Since $p$-flip models are strongly fragmented, such quantities still exhibit striking signatures of Hilbert space fragmentation. However, it is pertinent to ask whether there is new physics to be found when averaging over a more restricted set of states. Similar to the "freezing transition" between weak and strong shattering studied in Refs. [26, 38, 89], is it possible for different symmetry sectors to exhibit qualitatively distinct localization properties, perhaps with operator localization in the bulk?

Finally, can the methods presented herein find uses in other applications? Reference [59] showed that the pair-flip model tuned to its Rokhsar-Kivelson point [90,91] exhibits a ground-state half-chain entanglement entropy that scales as $\sqrt{L}$, with an inverse-polynomial scaling of the gap to the first-excited state (see also Ref. [62]). The generating functions presented in Eq. (26) can likely be used to make similar statements about the triple-flip model tuned to an equivalent frustration-free point in parameter space.

Most broadly, our results highlight the importance of exact results in physics. In this instance, numerical simulations were unable to observe the crossover from $1/L$ to $1/\sqrt{L}$ scaling, with important physical consequences, which only manifests in systems of size greater than $O(50)$ sites. There are other areas of physics, such as the field of many-body localization, where the extent to which finite-size numerics can be used to reliably infer behavior in thermodynamically large systems is a source of much debate. While the results presented herein cannot be applied directly to that problem, they do provide an interesting illustration of the fact that numerics on systems of finite size can be misleading, and there is no substitute for an exact result.

## Acknowledgments

I am grateful to Rahul Nandkishore, Charles Stahl, and David T. Stephen for fruitful discussions, and for previous collaboration on related topics. I would also like to thank them, along with Aaron Friedman and Alexey Khudorozhkov, for useful feedback on the manuscript. I also thank Paul Romatschke and Ted Nzuonkwelle for providing access to the Eridanus cluster at CU Boulder, on which the numerical simulations were performed.

**Funding information.** This work was supported by the Air Force Office of Scientific Research under Award No. FA9550-20-1-0222.

## A   Conserved quantities in $p$-flip dynamics

Here, we show that the operators in (10) are conserved by $p$-flip dynamics, and derive some simple relationships between conserved quantities specified by irreducible labels containing different numbers of colors. Since the quantities are conserved for all choices of the coefficients $g_i^{\alpha\beta}$ entering (2), they must commute with each term individually. Specifically, defining the local "kinetic" term $\hat{T}_i^{\alpha\beta} \equiv |\alpha\alpha\cdots\alpha\rangle\langle\beta\beta\cdots\beta|_{[i,i+p)}$, we wish to show that $[\hat{N}^{\alpha_1\alpha_2\cdots\alpha_k}, \hat{T}_i^{\alpha\beta}] = 0$ for all $i, \alpha, \beta$ only if the sequence of colors $\alpha_1\alpha_2\cdots\alpha_k$ forms a valid irreducible label for $p$-flip dynamics on a system of size $k$ with open boundaries. The commutator of interest is equal to

$$[\hat{N}^{\alpha_1\alpha_2\cdots\alpha_k}, \hat{T}_i^{\alpha\beta}] = \sum_{j_1<j_2<\cdots<j_k=0}^{L-1} \exp\left(\frac{2\pi i}{p}\sum_{n=1}^{k} j_n\right)\left[\prod_{n=1}^{k} \hat{P}_{j_n}^{\alpha_n}, \hat{T}_i^{\alpha\beta}\right]. \tag{A.1}$$

Since $\hat{T}_i^{\alpha\beta}$ only acts nontrivially on the interval $[i, i+p)$, only those $n$ such that $j_n \in [i, i+p)$ can possibly contribute to the commutator in the summand on the right-hand side of (A.1). Defining $I = \{n \mid j_n \in [i, i+p)\}$,

$$[\hat{N}^{\alpha_1\alpha_2\cdots\alpha_k}, \hat{T}_i^{\alpha\beta}] = \sum_{j_1<j_2<\cdots<j_k=0}^{L-1} \exp\left(\frac{2\pi i}{p}\sum_{n=1}^{k} j_n\right)\left[\prod_{n\in I} \hat{P}_{j_n}^{\alpha_n}, \hat{T}_i^{\alpha\beta}\right]. \tag{A.2}$$

Since $T_i^{\alpha\beta}$ annihilates configurations in which not all spins are equal to one another, the summand vanishes if $\{\alpha_{n\in I}\}$ are not all equal. Next, consider the case where all $\{\alpha_{n\in I}\}$ are equal to each other, and $|\{\alpha_{n\in I}\}| \equiv r < p$. In this case, the commutator is nonzero if the $\alpha_{n\in I}$ are equal to $\alpha$ or $\beta$. We must therefore look to the phases in order to obtain a cancellation. Since the value of the commutator does not depend on the particular configuration of projectors in the interval $[i, i+p)$ (and neither does the summation over all $j_{n\notin I}$ in the exterior regions), we can factor out a sum of the phases over the locations of the projectors, with each configuration of projectors in the interval appearing with equal weight:

$$\sum_{i_1<i_2<\cdots<i_r=0}^{p-1} \exp\left[\frac{2\pi i}{p}(i_1 + i_2 + \cdots + i_r)\right] = 0, \tag{A.3}$$

for $r < p$, where the the positions $i_n$ are offset from the $j_n$ by $i$. Hence, in order to obtain a nonzero commutator, it is necessary to consider a sequence of colors that contains at least one run of length $p$ (or greater). In this case, there exist configurations of projectors such that $|\{\alpha_{n\in I}\}| = p$, for which (A.3) no longer vanishes. However, the absence of such runs of length $p$ (or greater) is precisely the condition that defines a valid irreducible label. We have therefore shown that all sequences of colors of length $k$ that form a valid irreducible label will be conserved by $p$-flip dynamics.

Now consider operators specified by strings of colors of different lengths. For, say, triple-flip dynamics in a system of size $L$, there are no irreducible labels of size $L - 1$ or $L - 2$, since removing neighboring runs of length three will remove at least *three* spins from the spin pattern. It is therefore reasonable to expect that, for certain $k$, conserved quantities specified by an irreducible label of length $k$ can be written in terms of those specified by a label of length $k + 1$. By utilizing the sum

$$\sum_{j=j_1}^{j_2} \omega_p^j = \frac{\omega_p^{j_2+1} - \omega_p^{j_1}}{\omega_p - 1}, \tag{A.4}$$

where $\omega_p = \exp\left(\frac{2\pi i}{p}\right)$, we are able construct the following telescoping summation, valid for sites with indices $\{j_1, \dots, j_k\}$ satisfying $j_1 < j_2 < \cdots < j_k$

$$\sum_{j=0}^{j_1-1} \omega_p^j + \omega^{-1} \sum_{j=j_1+1}^{j_2-1} \omega_p^j + \omega^{-2} \sum_{j=j_2+1}^{j_3-1} \omega_p^j + \cdots + \omega^{-k} \sum_{j=j_k+1}^{L-1} \omega_p^j = \frac{\omega_p^{L-k}-1}{\omega_p-1}, \qquad \text{(A.5)}$$

which vanishes for $k = L \mod p$. If $k$ does not satisfy this condition, we can use the left-hand-side of (A.5) to resolve the identity. Specifically, a $k$-index conserved operator satisfies

$$\hat{N}^{\alpha_1 \alpha_2 \cdots \alpha_k} = \sum_{j_1 < j_2 < \cdots < j_k = 0}^{L-1} \frac{\omega_p - 1}{\omega_p^{L-k}-1} \left( \sum_{n=0}^{k} \omega^{-n} \sum_{j_n < j < j_{n+1}} \omega_p^j \right) \omega_p^{\sum_{n=1}^{k} j_n} \prod_{n=1}^{k} \hat{P}_{j_n}^{\alpha_n} \qquad \text{(A.6)}$$

$$= \frac{\omega_p - 1}{\omega_p^{L-k}-1} \sum_{\beta=0}^{m-1} \left( \hat{N}^{\beta \alpha_1 \alpha_2 \cdots \alpha_k} + \omega_p^{-1} \hat{N}^{\alpha_1 \beta \alpha_2 \cdots \alpha_k} + \cdots + \omega_p^{-k} \hat{N}^{\alpha_1 \alpha_2 \cdots \alpha_k \beta} \right), \qquad \text{(A.7)}$$

where, in the second line, we inserted $\sum_{\beta=0}^{m-1} \hat{P}_j^{\beta} = \mathbb{1}$ in each term in the sum. For $k \neq L \mod p$, this expression relates a $k$-index conserved operator to a linear combination of $(k+1)$-index operators. Note, however, that, while the sequence of colors $\{\alpha_1, \alpha_2, \dots, \alpha_k\}$ forms a valid irreducible label, the operators on the right-hand side of (A.7) may involve nonconserved operators. In particular, if the sequence $\{\alpha_1, \alpha_2, \dots, \alpha_k\}$ contains a run of color $\alpha$ of length $p-1$, then inserting the color $\alpha$ before, after, or within this run will generate a run of length $p$. However, the resulting nonconserved operator appears in (A.7) multiplied by $1 + \omega^{-1} + \cdots + \omega^{-(p-1)} = 0$, implying that only valid irreducible labels contribute to the right-hand side of (A.7). We have therefore shown that, via (A.7), $k$-index conserved operators can be written in terms of $(k+1)$-index conserved operators (for $k \neq L \mod p$). This relationship substantially reduces the number of linearly independent conserved operators, but there remains some residual redundancy; for each irreducible label, Eq. (10) has both real and imaginary parts. Constructing another minimal (i.e., linearly independent) basis of conserved operators from (10), in addition to the projectors onto Krylov sectors already constructed in the main text, lies beyond the scope of this work.

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
