# Peer review of "Exact Mazur bounds in the pair-flip model and beyond"

_SciPost Physics Core, doi:SciPost Phys. Core 7, 040 (2024)_

## Round 1 · Referee Report · Anonymous (Referee 1) · 2023-10-6

Strengths

1- computation of Mazur bounds that were previously unavailable

Weaknesses

1- Physical motivation for the model lacking 2- Relevant boundary charges are local

Report

The author computes local and non-local charges for a "p-flip" spin model previously studied numerically in Ref. [18] and uses them compute Mazur bounds for certain \emph{local} observables. The ones close to the boundary are finite in the thermodynamic limit L \to \infty and the ones in the bulk decay as 1/\sqrt{L}. However, the author refers to them as non-local, but if they give finite Mazur bounds for strictly local observables, they must be local operators and indeed there seem to be conservation laws localized near the boundary by inspection of Eq. (10).

Requested changes

1- Discuss as above local vs nonlocal more precisely 2- More physical motivation for the model is needed beyond that it has been studied in the previous literature

  • validity: high
  • significance: ok
  • originality: good
  • clarity: high
  • formatting: excellent
  • grammar: perfect

Author:  Oliver Hart  on 2024-04-04  [id 4389]

(in reply to Report 1 on 2023-10-06)

1- Discuss as above local vs nonlocal more precisely

In what follows, I use the updated numbering of equations, citations, etc that appear in the resubmitted version of the manuscript, although I will highlight where numbers have changed.

One can indeed write down a more local set of conserved quantities that likely generate the conserved quantities in Eq. (10) and their construction is intimately related to the data presented in Fig. 3. These are the "SLIOMs" (statistically localized integrals of motion) of Ref. [20] (previously [14]). A special case is the color of the leftmost or rightmost dots in the irreducible label, which defines a conserved quantity ``localized'' near the left and right boundaries, respectively.

The discussion below focuses on the pair-flip model, but equivalent statements can be made for the triple-flip model. Recall that the irreducible label - obtained by decimating neighboring pairs of spins with the same color - is composed of "dots," and each irreducible label is in one-to-one correspondence with a Krylov sector. Consider the Mazur bound that arises from conservation of the color of, say, the leftmost dot in the label. The relevant conserved operator is

$$\hat{I}0 = \sum_j \hat{P}^z_j $$}\hat{S
where $\hat{P}_{\mu j}$ is a projector, diagonal in the computational basis, that projects onto states in which the $\mu$th dot in the label, as measured from the left boundary, appears on site $j$. The operator $\hat{S}^z_j$ was defined in the manuscript as $\hat{S}^z_j = \sum_\alpha (m-1-2\alpha)/2 |\alpha\rangle\langle\alpha|$. The precise sense in which the operator $\hat{I}_0$ is localized in the vicinity of the left boundary is described in detail in Ref. [20] and also in the new Ref. [61], which defines such quantities as "cryptolocalized," since they cannot be expressed in terms of strictly local operators.

The operator $\hat{I}_0$ is challenging to make analytical progress with since it contains projection operators that implicitly involve the decimation procedure. Nevertheless, we can evaluate the Mazur bound using $\hat{I}_0$ for a particular choice of decimation procedure. We need $(S^z_0 | I_0)$ and $(I_0 | I_0)$ to evaluate the bound. Orthogonality of projectors ensures that $(I_0 | I_0) = D^{-1} \sum_j \mathrm{Tr}[P_{0j} (S^z_j)^2]$, while the former becomes $(S^z_0 | I_0) = D^{-1} \sum_j \mathrm{Tr}(P_{0j} S^z_0 S^z_j)$. Whether $S^z_0$ and $S^z_j$ are orthogonal in the subspace defined by $P_{0j}$ depends on the specific decimation procedure used. The following ensures that they are orthogonal: look for pairs from left to right, then, if a pair is found, delete it and move back to the left edge. Equipped with this choice of decimation procedure, the bound reduces to

$$M_0 = \frac{(\mathrm{Tr} P_{0,0} Z_0^2)^2}{D \sum_j \mathrm{Tr} P_{0,j} Z_j^2} = \frac{1}{12D}(m^2-1) \frac{(\mathrm{Tr} P_{0,0})^2 }{\sum_j \mathrm{Tr} P_{0,j} } = \langle S_0(0) S_0(0) \rangle_{\beta=\infty} \left[ \frac{\mathrm{Tr} P_{0,0} }{\sum_j \mathrm{Tr} P_{0,j} } \right]^2 [1 + o(1)]$$
where we used the fact that, in the limit of large systems, the fraction of states for which there is no first dot in the label vanishes exponentially with $L$. The term in the square brackets is the probability of finding the first dot in the label on the first site. This is what is plotted in the left panel of Fig. 3 (although using a different decimation procedure), which shows that the probability is nonzero, leading to $M_0 > 0$. This bound is notably not tight - additional conserved quantities must be included to obtain a tight bound. However, the operator $\hat{I}_0$ is sufficient to establish nonzero autocorrelation at the boundary. It is also important to note that, while the decomposition of the bound in terms of projectors depends on the choice of decimation procedure, the numerical value of the bound does not.

While it is conceptually useful to write down such pseudolocalized conserved operators in order to provide intuition for the infinite autocorrelation times observed at the boundary, the principal goal of the manuscript was to provide tight Mazur bounds. As shown in Ref. [24] (previously [18]), inlcuding only the SLIOMs does not give rise to a tight bound throughout the bulk and it is conventient to use the full set of operators in (10). Note that we are interested in Mazur bounds that vanish with system size. Furthermore, taking into account a larger number of SLIOMs becomes analytically challenging.

In summary, I have clarified in the main text that there exist conserved quantities that are more local than the complete basis of operators presented in Eq. (10), and that these conserved operators can be used to obtain an intuitive picture of the infinite autocorrelation time for strictly local operators at the boundary.

2- More physical motivation for the model is needed beyond that it has been studied in the previous literature

I thank the referee for this comment. However, I disagree that the motivation for studying the model given in the manuscript is insufficient. I strongly believe that its status as one of the canonical models of fragmentation (i.e., appearing frequently in the literature) is sufficient to motivate understanding its behavior better, especially given that this was a point of confusion. Indeed, at least from my perspective, the primary purpose of the model is to elucidate the possible phenomena that fragmented models may exhibit, as opposed to any particular more "physical" motivation.

Nevertheless, it is relevant to the physics of the following more well-known models/concepts:

  • As noted in Ref. [59] (previously [47]), the $m=2$ pair-flip model tuned to its Rokhsar-Kivelson (RK) point (aka the Temperley-Lieb model, which exhibits at least as many conserved quantities as the pair-flip model) is unitarily equivalent to the spin-$1/2$ Heisenberg model up to a trivial energy shift.
  • Similarly, as noted in Ref. [24] (previously [18]), the $m=3$ Temperly-Lieb model is unitarily equivalent to $\mathrm{SU}(3)$-symmetric spin-1 Hamiltonians of the form $\sum_j J_j [(S_j \cdot S_{j+1})^2 - 1]$.
  • As noted in Ref. [47] (previously [41]), the pair-flip model for generic $m$ is (Kramers-Wannier) dual to a parity-sensitive PXP model in which a spin is only dynamical if its two neighbors are the same color, connecting the physics of Eq. (2) to kinetically constrained models with facilitated spin flips.
  • Many fragmented models derive their properties from a conserved pattern. The methods presented in the present manuscript may be used to quantitatively understand their dynamics.

Some of these connections are now mentioned explicitly in the manuscript.

---

## Round 1 · Referee Report · Anonymous (Referee 2) · 2023-11-14

Report

The Author performs an original analytical calculation of Mazur bounds on correlation functions in a class of models exhibiting Hilbert-space fragmentation. The results point out a discrepancy between the scalings of boundary and bulk correlations and their decay, and they show that a previous numerics-based conjecture regarding the scalings is incorrect and, in fact, currently available numerical resources are sufficient to establish such a conjecture. This work is a nice addition to the literature tackling the non-thermal behaviour in fragmented models with original, exact analytical methods.

The paper is written rather technically and, in my opinion, addresses an interesting specific problem in the field of Hilbert-space fragmented models. It certainly warrants publication in some form. However, I have checked the criteria for acceptance in SciPost Physics and am unfortunately not convinced that any one of the required are satisfied:

  • Detail a groundbreaking theoretical/experimental/computational discovery;

  • Present a breakthrough on a previously-identified and long-standing research stumbling block;

  • Open a new pathway in an existing or a new research direction, with clear potential for multipronged follow-up work;

  • Provide a novel and synergetic link between different research areas.

I should stress that these criteria are strict so my judgement does not diminish the relevance of the paper in solving a particular question of a specific community. In fact, I found the methods used quite intriguing and believe they have an above-the-norm degree of originality. I admire in particular how the Author manages to point out a peculiar crossover between two different scaling behaviours, which is extremely hard to observe with currently accessible numerical resources. This work thus certainly could be published in SciPost Physics Core according to its required criteria! I would recommend publication there, after remarks below have been addressed.

The main criticism of mine concerns the role of locality for thermalization or lack thereof [see remark (5) below]. In particular, the Author regularly emphasizes the role played by non-local conserved labels of Hilbert space fragments in the lack of thermalization. However, the redundacny of such objects is never really checked, since bounds on correlation functions are computed using the entire basis of projectors on Krylov sectors. I do not see how this excludes possible existence of local conserved quantities that would saturate the Mazur bound. Specifically, since there is a clear discrepancy between behaviours of the bulk and the boundary correlators (a discrepancy which, as pointed out by the Author, vanishes in a periodic chain described by a model with a similar phenomenology), one should perhaps exclude the possibility of boundary-localized modes preventing thermalization before claiming the role of non-local conserved quantities in that respect. Additional critique is regarding acknowledgement of other works. I would suggest the Author to consider checking more of the literature that relates the concepts of Mazur bounds, locality, and thermalization. Please find a short non-exhaustive list of recommendations in points (6) and (7) below.

Requested changes

(1) The Introduction is occasionally a bit cryptic. For example, the Author states “It is now understood that Hilbert space fragmentation can arise in more generic settings from the simultaneous enforcement of symmetries and strict spatial locality …” — What exactly is meant by simultaneous enforcement of symmetries and strict spatial locality? Locality of what?

(2) Typo on page 3: “This suggests that, in the bulk, the nonlocal patterns conserved by the dynamics do not change the scaling that [one?] obtains from the local continuous symmetries of the pair-flip model.”

(3) Between eqs. (14), (15), please define $\hat{S}^z(\alpha)$. (I guess it is the diagonal element of the $\hat{S}^z$.)

(4) If I understand correctly the discussion of labels of Krylov sectors in Section 2.2, the patterns that remain after a decimation can always be moved to one side of the system, say, left. If one then cuts that left part of the system containing only the label away, that cut part is frozen. On the right side of the system, on the other hand, remain only the contiguous sequences of colours that can be flipped and reshuffled. This reminds of a typical situation of Hilbert-space fragmented systems, where any Krylov sector can be constructed by joining a frozen configuration of a subsystem (that frozen configuration determines the label of the Krylov sector) to a vacuum, or some other state containing movable particles, on the rest of the system — see for instance reference [16]. Perhaps the Author could comment on this analogy, if correct.

(5) The Author repeatedly refers to the “effect of non-local conserved quantities” [i.e., the ones in eq. (10)] on the dynamics. While the Mazur bounds are calculated using a basis that is equivalent to the non-local operators in eq. (10), it is still not clear whether and how many of these quantities are redundant. In other words, I am not sure that the calculations presented herein exclude the possibility that there exist some (perhaps small set of) local objects that are conserved and prevent thermalization. Especially so, since there is a discrepancy between the bulk and the boundary scaling of the correlation functions. — Since this discrepancy vanishes in the t-J model with periodic boundary conditions, should one expect some kind of a boundary-localized operator [e.g., à la J. Stat. Mech. (2017) 063105] to be behind the slowed-down thermalization or non-thermalization of the correlators? While this question probably cannot be answered in this work, I would perhaps suggest the Author to at least refrain from statements that would imply non-local quantities affecting the correlation functions of local observables, as their redundancy has not been checked. Perhaps a more neutral statement that “fragmentation” affects thermalization is more proper, although its message is not so surprising.

(6) Since the alleged effect of non-local conservation laws is emphasized, it would be fitting to mention the fundamental role of locality/extensivity of conservation laws that affect thermalization — see, e.g., refs. Commun. Math. Phys. 351, 155 (2017); Phys. Rev. Lett. 115, 157201 (2015); J. Stat. Mech. (2016) 064008, etc.

(7) I believe that more works should be acknowledged, in particular concerning Mazur bounds and on the concepts of thermalization and correlation functions. Some suggestions:

  • Mazur bounds on transport coefficients: Phys. Rev. B 55, 11029 (1997); Phys. Rev. Lett. 106, 217206 (2011); Phys. Rev. B 83, 035115 (2011); Commun. Math. Phys. 318, 809 (2013); Phys. Rev. Lett. 111, 057203 (2013); Nucl. Phys. B 886, 1177 (2014); Phys. Rev. Lett. 122, 150605 (2019).

  • Some references on hydrodynamic projections (Mazur bounds can be thought of as truncations of the latter): SciPost Phys. 3, 039 (2017); J. Stat. Phys. 186, 25 (2022); Commun. Math. Phys. 391, 293–356 (2022).

  • Mazur bound used in establishing non-thermal behaviour of correlation functions: Phys. Rev. B 102, 041117(R); SciPost Phys. 9, 003 (2020).

  • Some references concerning lack of thermalization signalled by slow correlation decay in structural glasses and related models: Phys. Rev. X 10, 021051 (2020); Phys. Rev. B 92, 100305(R) (2015); Phys. Rev. Lett. 121, 040603 (2018); Phys. Rev. B 108, L100304, (2023); arXiv:2306.12467 (2023).

  • Lack of thermalization of boundary spins due to edge modes: J. Stat. Mech. (2017) 063105, and refs. therein.

(8) Enumeration of equations needs to be fixed — see eqs. (38) and (52) and neighbouring ones.

(9) Section 4.1.2, the Author states “… only the first (last) dot in the label is able to touch the leftmost (rightmost) site under p-flip dynamics. Hence, the fact that nonzero autocorrelation is able to develop follows from the nonzero “return probability” for the first and last dots in the label.” I do not see the implication from the first to the second sentence. Perhaps this should be explained in an additional sentence/figure, or, if trivial, rephrased so that it becomes obvious.

(10) Related to the previous point: Section 4.3.1 uses the same implication. I would appreciate some explanation, to make the rather technical part of the work easier to read.

(11) Could more numerics be performed, independently of the analytical methods used herein, in order to benchmark some of the steps of the analytical calculation (to make a reader who wishes to go through technical details fast trust the analytical calculation)?

(12) Is there any physical reason or intuitive explanation for the crossover between the two scaling behaciours in fig. 6?

  • validity: ok
  • significance: ok
  • originality: high
  • clarity: ok
  • formatting: good
  • grammar: excellent

Author:  Oliver Hart  on 2024-04-04  [id 4390]

(in reply to Report 2 on 2023-11-14)

I should stress that these criteria are strict so my judgement does not diminish the relevance of the paper in solving a particular question of a specific community. In fact, I found the methods used quite intriguing and believe they have an above-the-norm degree of originality. I admire in particular how the Author manages to point out a peculiar crossover between two different scaling behaviours, which is extremely hard to observe with currently accessible numerical resources. This work thus certainly could be published in SciPost Physics Core according to its required criteria! I would recommend publication there, after remarks below have been addressed.

I thank the referee for the overall positive assessment. Based on the recommendation of the referee, I have decided to resubmit the manuscript to SciPost Physics Core.

The main criticism of mine concerns the role of locality for thermalization or lack thereof [see remark (5) below]. In particular, the Author regularly emphasizes the role played by non-local conserved labels of Hilbert space fragments in the lack of thermalization. However, the redundacny of such objects is never really checked, since bounds on correlation functions are computed using the entire basis of projectors on Krylov sectors. I do not see how this excludes possible existence of local conserved quantities that would saturate the Mazur bound. Specifically, since there is a clear discrepancy between behaviours of the bulk and the boundary correlators (a discrepancy which, as pointed out by the Author, vanishes in a periodic chain described by a model with a similar phenomenology), one should perhaps exclude the possibility of boundary-localized modes preventing thermalization before claiming the role of non-local conserved quantities in that respect. Additional critique is regarding acknowledgement of other works. I would suggest the Author to consider checking more of the literature that relates the concepts of Mazur bounds, locality, and thermalization. Please find a short non-exhaustive list of recommendations in points (6) and (7) below.

I did not mean to imply that it is impossible to construct conserved quantities with increased locality from the family of operators in Eq. (10). My use of the term "nonlocal" was intended to distinguish the simple global symmetries of the model (discussed in Sec. 2.1) from "the rest" of the conserved quantities implied by conservation of the irreducible label. In the updated manuscript, I explicitly construct a single conserved quantity that provides a nonzero Mazur bound at the edge of the system. This operator cannot be expressed as a sum of strictly local operators, but is nevertheless "pseudolocalized" with respect to the infinite-temperature state. See below and also my response to Referee 1.

The Introduction is occasionally a bit cryptic. For example, the Author states “It is now understood that Hilbert space fragmentation can arise in more generic settings from the simultaneous enforcement of symmetries and strict spatial locality …” — What exactly is meant by simultaneous enforcement of symmetries and strict spatial locality? Locality of what?

Hilbert space fragmentation typically arises when one looks at the most local Hamiltonian consistent with a given set of symmetries. For example, one may regard the pair-flip Hamiltonian given in Eq. (2) as the most local Hamiltonian (i.e., acting on the fewest contiguous sites) that is consistent with the symmetries in Eq. (1). If we allow terms with larger range or support, then there are many other Hamiltonian contributions compatible with Eq. (1) (notably, at the next-nearest-neighbor level, colors can be freely exchanged). I have reworded this sentence in the introduction to make this point clearer.

Typo on page 3: “This suggests that, in the bulk, the nonlocal patterns conserved by the dynamics do not change the scaling that [one?] obtains from the local continuous symmetries of the pair-flip model.”

I thank the referee for spotting this. The typo has been fixed.

Between eqs. (14), (15), please define $S^z(\alpha)$. (I guess it is the diagonal element of the $S^z$.)

The referee is indeed correct. The coefficients $S^z(\alpha)$ are the diagonal matrix elements of $S^z$ in the basis $|\alpha\rangle$. The explicit expression for these coefficients has now been included in the manuscript between Eqs. (14), (15).

If I understand correctly the discussion of labels of Krylov sectors in Section 2.2, the patterns that remain after a decimation can always be moved to one side of the system, say, left. If one then cuts that left part of the system containing only the label away, that cut part is frozen. On the right side of the system, on the other hand, remain only the contiguous sequences of colours that can be flipped and reshuffled. This reminds of a typical situation of Hilbert-space fragmented systems, where any Krylov sector can be constructed by joining a frozen configuration of a subsystem (that frozen configuration determines the label of the Krylov sector) to a vacuum, or some other state containing movable particles, on the rest of the system — see for instance reference [16]. Perhaps the Author could comment on this analogy, if correct.

This is correct. As I outline in the manuscript, all states that share the same irreducible label belong to the same Krylov sector and can therefore be connected by local pair-flip moves. This includes the state where the label is pushed up against one boundary and the remainder of the system is filled with a uniform color or, more generally, a noncrossing dimer configuration (neither of which contributes to the label). The $t$-$J_z$ model, mentioned throughout the manuscript, is another model where Krylov sectors are identified by a conserved pattern that follows from performing a decimation procedure (in that case, removing all empty sites). The existence of an underlying "pattern" conserved by the dynamics defines a class of fragmented systems. Note that Ref. [16] has become Ref. [22] in the updated manuscript.

There are some important similarities between the model of Ref. [22] and the pair-flip model. First, for spin-$1/2$ degrees of freedom, the pair-flip model in Eq. (2) is Kramers-Wannier dual to a model of the form $P_{i-1} \sigma_i^x P_{i+1}$, where the projection is onto states where the spins on sites $i-1$ and $i+1$ match (i.e., configurations of the form $0 \bullet 0$ and $1 \bullet 1$ are dynamical), with natural generalizations to $m>2$. This is to be compared with the Hamiltonian from Ref. [22], which is of the form $P_{i-1} \sigma_i^+ \sigma_{i+1}^- P_{i+2} + \text{h.c.}$, so that $ \sigma_i^+ \sigma_{i+1}^-$ only acts if the spins on either side match (which follows from conservation of domain wall number). Both models therefore exhibit flipping or hopping facilitated by the same constraint. Second, one can view their construction in the "decimation procedure" perspective taken in the present manuscript. In the domain wall language (now denoting the presence/absence of a domain wall by 1/0), Ref. [22]'s Hamiltonian sends $011 \leftrightarrow 110$, i.e., isolated magnons are mobile. However, this dynamics leaves a pattern unchanged. Namely, one first finds the domain wall configuration that corresponds to the spin configuration. Next, one removes all neighboring 1's. This naturally leads to irreducible labels satisfying the "Fibonacci" constraint (no neighboring 1's). Each such label identifies a unique Krylov sector. The existence of the conserved pattern means it is plausible that much of the phenomenology discussed in the present manuscript also applies to the dynamics exhibited by the model of Ref. [22] at infinite temperature.

The Author repeatedly refers to the “effect of non-local conserved quantities” [i.e., the ones in eq. (10)] on the dynamics. While the Mazur bounds are calculated using a basis that is equivalent to the non-local operators in eq. (10), it is still not clear whether and how many of these quantities are redundant. In other words, I am not sure that the calculations presented herein exclude the possibility that there exist some (perhaps small set of) local objects that are conserved and prevent thermalization. Especially so, since there is a discrepancy between the bulk and the boundary scaling of the correlation functions. — Since this discrepancy vanishes in the t-J model with periodic boundary conditions, should one expect some kind of a boundary-localized operator [e.g., à la J. Stat. Mech. (2017) 063105] to be behind the slowed-down thermalization or non-thermalization of the correlators? While this question probably cannot be answered in this work, I would perhaps suggest the Author to at least refrain from statements that would imply non-local quantities affecting the correlation functions of local observables, as their redundancy has not been checked. Perhaps a more neutral statement that “fragmentation” affects thermalization is more proper, although its message is not so surprising.

I thank the referee for allowing me to clarify this very important point. There is also substantial overlap with the principal criticism of Referee 1. As pointed out in Ref. [24] (previously [18]), it is expected that the conserved quantities in Eq. (10) are generated by a smaller set of conserved quantities known as "SLIOMs" (statistically localized integrals of motion) introduced in Ref. [20] (previously [14]). These are constructed by writing down an operator that corresponds to the color of the $\mu$th dot in the irreducible label. I show in the updated manuscript that the color of, e.g., the leftmost dot is sufficient to establish a nonzero Mazur bound for left-boundary autocorrelation functions. The connection between boundary SLIOMs and strong zero modes was already pointed out and discussed in Ref. [20].

I would however like to stress that the principal goal of the present manuscript is to obtain tight analytical Mazur bounds throughout the bulk, and this requires taking into account more than just the SLIOMs, even if they generate all conserved quantities in Eq. (10)! Indeed, it has been shown in Ref. [24] that, for the $t$-$J_z$ model [where the SLIOMs provably generate the full set of conserved quantities analogous to Eq. (10)], using the entire set of SLIOMs does not lead to tight Mazur bounds in the bulk (see, e.g., Fig. 3 of Ref. [24]). It is therefore convenient to use the full set of operators in Eq. (10).

The question of whether there exist underlying boundary-localized operators "responsible" for the breakdown of thermalization is therefore now thoroughly addressed in the manuscript. In terms of language, I have opted to say that it is "the conserved pattern" that affects thermalization.

Since the alleged effect of non-local conservation laws is emphasized, it would be fitting to mention the fundamental role of locality/extensivity of conservation laws that affect thermalization — see, e.g., refs. Commun. Math. Phys. 351, 155 (2017); Phys. Rev. Lett. 115, 157201 (2015); J. Stat. Mech. (2016) 064008, etc.

I thank the referee for these suggestions. The referencing in the manuscript has been updated accordingly.

I believe that more works should be acknowledged, in particular concerning Mazur bounds and on the concepts of thermalization and correlation functions. Some suggestions:

  • Mazur bounds on transport coefficients: Phys. Rev. B 55, 11029 (1997); Phys. Rev. Lett. 106, 217206 (2011); Phys. Rev. B 83, 035115 (2011); Commun. Math. Phys. 318, 809 (2013); Phys. Rev. Lett. 111, 057203 (2013); Nucl. Phys. B 886, 1177 (2014); Phys. Rev. Lett. 122, 150605 (2019).
  • Some references on hydrodynamic projections (Mazur bounds can be thought of as truncations of the latter): SciPost Phys. 3, 039 (2017); J. Stat. Phys. 186, 25 (2022); Commun. Math. Phys. 391, 293–356 (2022).
  • Mazur bound used in establishing non-thermal behaviour of correlation functions: Phys. Rev. B 102, 041117(R); SciPost Phys. 9, 003 (2020).
  • Some references concerning lack of thermalization signalled by slow correlation decay in structural glasses and related models: Phys. Rev. X 10, 021051 (2020); Phys. Rev. B 92, 100305(R) (2015); Phys. Rev. Lett. 121, 040603 (2018); Phys. Rev. B 108, L100304, (2023); arXiv:2306.12467 (2023).
  • Lack of thermalization of boundary spins due to edge modes: J. Stat. Mech. (2017) 063105, and refs. therein.

I thank the referee for these suggestions. The referencing in the manuscript has been updated accordingly.

Enumeration of equations needs to be fixed — see eqs. (38) and (52) and neighbouring ones.

This enumeration of equations was intentional. Since the equations are in the "summary of results" section, the equation numbers point to the location in the document where that equation is derived (they are hyperlinks). This is my preference, so I have decided to leave the numbering as it was. If, after this explanation, the referee still feels strongly that this choice is confusing, I am happy to change it.

Section 4.1.2, the Author states “… only the first (last) dot in the label is able to touch the leftmost (rightmost) site under p-flip dynamics. Hence, the fact that nonzero autocorrelation is able to develop follows from the nonzero “return probability” for the first and last dots in the label.” I do not see the implication from the first to the second sentence. Perhaps this should be explained in an additional sentence/figure, or, if trivial, rephrased so that it becomes obvious.

I thank the referee for allowing me to clarify this point. Consider expressing the trace in the autocorrelation function $\mathrm{Tr}[S_i^z(t)S_i^z(0)]$ in terms of $S^z$ eigenstates $|\alpha\rangle$. It reduces to $ \sum_\mu \sum_{{\alpha \in \mu}} \langle{\alpha}(t) | S^z_i | {\alpha}(t) \rangle S^z_i({\alpha}) $, where $\mu$ runs over Krylov sectors and $\alpha \in \mu$ are the states belonging to the sector $\mu$. If the system thermalizes within each Krylov sector, then we can replace $\langle{{\alpha}(t) | S^z_i | {\alpha}(t) }\rangle = d_{\mu}^{-1} \sum_{\beta \in \mu} \langle{\beta |S^z_i| \beta }\rangle$, with $d_\mu$ the dimension of the Krylov sector. Combining, we have

$$\frac{1}{D}\sum_\mu \frac{1}{d_\mu} \sum_{\alpha, \beta \in \mu} S^z_i({\alpha}) S^z_i({\beta})$$
which is just the Mazur bound in Eq. (18). The point of reproducing the Mazur bound in this way is to justify thinking about $\alpha$ as the "initial" state, and $\beta$ as the "final" state (which we average over).

The intuition behind the sentence is that nonzero autocorrelation is able to develop on site $i$ if, when the site is initially occupied by a dot (in the state $\alpha$), the dot has a nonzero probability of "returning" to site $i$ (over the ensemble of final states $\beta$). Otherwise, $S_i^z(\alpha)S_i^z(\beta)$ averages to zero, since only the irreducible label must be preserved. This argument is not quantitative because the position of the dots is not sharply defined, being a function of the decimation procedure. However, different decimation procedures only give rise to $O(1)$ differences in position, so the scaling with $L$ shouldn't be affected.

I have modified the sentence in the manuscript to make this logic easier to follow. The SLIOM bound presented in the new Sec. 4.1.4 places this intuitive argument on more rigorous footing by relating the probability that the leftmost dot is found on the leftmost site (for a special choice of decimation procedure) to a bound on autocorrelation functions.

Related to the previous point: Section 4.3.1 uses the same implication. I would appreciate some explanation, to make the rather technical part of the work easier to read.

The same reasoning is not used in Sec. 4.3.1. It is stated below Eq. (30) that the leftmost site of the lattice can only ever "see" the leftmost dot in the irreducible label, since the ordering of dots in the label is preserved. This is why Eq. (30) depends only on the color of the first dot in the label, while Eq. (44) depends on the entire irreducible label. Perhaps the referee meant 4.1.3, in which case I hope the updated explanation in Sec. 4.1.2 clarifies the matter.

Could more numerics be performed, independently of the analytical methods used herein, in order to benchmark some of the steps of the analytical calculation (to make a reader who wishes to go through technical details fast trust the analytical calculation)?

I have included numerical data in Figs. 4 and 6 corresponding to numerically exact evaluation of Eq. (18). This was performed in the following way: (i) finding the adjacency matrix $A_{\alpha\beta}$ corresponding to the Hamiltonian in Eq. (2) assuming all coefficients are nonzero, (ii) using $A_{\alpha\beta}$ to find the system's Krylov sectors in the computational basis $| \alpha \rangle$, (iii) evaluating the sum over computational basis states belonging to each Krylov sector. This constitutes an independent numerical check of the analytical manipulations presented in the manuscript, and the data match perfectly with the exact expressions in the manuscript.

Additionally, Appendix A of Ref. [60] (previously [48]) now includes updated numerics based on classical cellular automaton evolution, which are able observe the $1/L$ to $1/\sqrt{L}$ crossover, inspired by the present submission.

I hope that the above results will inspire additional faith in the validity of the exact analytical manipulations presented in the manuscript.

Is there any physical reason or intuitive explanation for the crossover between the two scaling behaciours in fig. 6?

It is likely related to the fact that the smallest lengthscale over which the position of a dot can be resolved is actually rather large for the pair-flip model, implying that large system sizes must be accessed in order to "see" that are really localized (even anomalously). To obtain the corresponding lengthscale, consider the left panel of Fig. 3. The probability that the first dot is found at position $\ell$ can be obtained by considering the number of noncrossing dimer configurations of length $\ell$, which scales asymptotically as $\sim[2\sqrt{m-1}]^\ell$ [set by the distance to the nearest singularity in the complex plane in Eq. (25) for $\psi_{2,m}(z)$], and comparing this to the total number of configurations, $m^\ell$. Hence, we expect asymptotic exponential decay of $P_\partial(\ell)$ into the bulk, up to polynomial corrections, with the lengthscale

$$\ell_m^{-1} = \log\frac{m}{2\sqrt{m-1}}$$
We have $\ell_3 \approx 39$ for the three-colored pair-flip model, which is approximately consistent with the crossover observed in Fig. 6. The same lengthscale controls differences in position with respect to different choices of decimation procedure.

I have added a sentence to Sec. 4.4.1 highlighting the existence of this large lengthscale in the problem.

---

## Round 2 · Referee Report · Anonymous (Referee 2) · 2024-5-11

Report

The Author has sufficiently answered all of the remarks/criticisms of the referees and has made the requested changes. I thank them for providing clear and concise explanations in their response to my concerns. I believe the manuscript is now perfectly suitable for publication in SciPost Physics Core, as resubmitted.

A small remark: regarding the model in reference [22], there in fact exists additional literature that could be of interest to the Author, independently of the current submission. It seems the model can be tackled using Bethe ansatz [J. Phys. A: Math. Gen. 24 L549 (1991), SciPost Phys. Core 4, 010 (2021), Phys. Rev. E 104, 044106 (2021)] and its fragmentation has been linked to special conserved quantities already in a classical setting [J. Stat. Phys. 86, 1237 (1997)].

Recommendation

Publish (meets expectations and criteria for this Journal)

---

## Round 2 · Referee Report · Anonymous (Referee 1) · 2024-5-14

Report

The author has fully addressed all my concerns and I recommend publication.

Recommendation

Publish (meets expectations and criteria for this Journal)

---

## Round 2 · Author Response

Dear Editors,

Thank you for arranging the review of this manuscript. Given the recommendation of the second referee, I am resubmitting the manuscript for consideration in SciPost Physics Core. I have left a detailed response to each referee on the submission page, which thoroughly addresses their comments and criticisms. The manuscript has been modified accordingly and I am confident that it will now be judged suitable for publication.

Sincerely,

Oliver Hart

---

## Round 2 · List of Changes

1. Added a new section (4.1.4) describing a boundary-localized operator that is sufficient to establish asymptotically nonzero autocorrelation functions at the edges of the system.
  2. Included numerical data in Figures 4 and 6 to verify the exact analytical manipulations presented in the manuscript.
  3. Removed all instances of the phrase "nonlocal conserved quantities" to avoid confusion.
  4. Improved referencing throughout the manuscript.
  5. Highlighted the existence of a large length scale in the problem, which provides some intuition for the crossover observed in Figure 6.

---

## Editorial Decision

published